

# Towards robust seasonal streamflow forecasts in mountainous catchments: impact of calibration metric selection in hydrological modeling

Diego Araya[1], Pablo A. Mendoza[1,2], Eduardo Muñoz-Castro[1] and James McPhee[1,2]

[1]Department of Civil Engineering, Universidad de Chile, Santiago, Chile
[2]Advanced Mining Technology Center, Universidad de Chile, Santiago, Chile

*Correspondence to*: Pablo A. Mendoza (pamendoz@uchile.cl)

**Abstract.** Dynamical (i.e., model-based) methods are widely used by forecasting centers to generate seasonal streamflow forecasts, building upon process-based hydrological models that require parameter specification (i.e., calibration). Here, we investigate the extent to which the choice of calibration objective function affects the quality of seasonal (spring-summer) streamflow forecasts produced with the traditional ensemble streamflow prediction (ESP) method and explore connections between forecast skill and hydrological consistency - measured in terms of biases in hydrological signatures - obtained from the model parameter sets. To this end, we calibrate three popular conceptual rainfall-runoff models (GR4J, TUW, and Sacramento) using 12 different objective functions, including seasonal metrics that emphasize errors during the snowmelt period, and produce hindcasts for five initialization times over a 33-year period (April/1987 – March/2020) in 22 mountain catchments that span diverse hydroclimatic conditions along the semiarid Andes Cordillera (28°-37°S). The results show that the choice of calibration metric becomes relevant as the winter (snow accumulation) season begins (i.e., July 1), enhancing inter-basin differences in forecast skill as initializations approach the beginning of the snowmelt season (i.e., September 1). The comparison of seasonal forecasts obtained from different calibration metrics shows that hydrological consistency does not ensure satisfactory seasonal ESP forecasts (e.g., Split KGE), and that satisfactory ESP forecasts are not necessarily associated to a hydrologically consistent parameter set (e.g., VE-Sep). Among the options explored here, an objective function that combines the Kling-Gupta Efficiency (KGE) and the Nash-Sutcliffe Efficiency (NSE) with flows in log space provides the best compromise between hydrologically consistent model simulations and good forecast performance. Finally, the choice of calibration metric generally affects the magnitude of correlations between forecast quality attributes and catchment descriptors, rather than the sign, being the baseflow index and interannual runoff variability the best predictors of forecast skill. Overall, this study highlights the need for careful parameter estimation strategies in the forecasting production chain to generate skillful forecasts for the right reasons and draw robust conclusions on hydrologic predictability.



## 1 Introduction

Seasonal streamflow forecasts can support long-term water resources management and planning, including allocations for water supply, irrigation, hydropower generation, industry, mining operations, and navigation. Therefore, improving the quality of these products is an ongoing challenge for the hydrology community, especially in regions where drought risk and severity are expected to increase under climate change scenarios (Cook et al., 2022). Among the existing approaches, dynamical methods – which rely on the implementation of hydrological or land surface models (Wood et al., 2018; Troin et al., 2021;

Slater et al., 2022) – are attractive because they involve explicit hydrologic process representations, with varying degrees of abstraction depending on model complexity (Hrachowitz and Clark, 2017). Accordingly, dynamical systems not only offer the opportunity to monitor and predict other variables than streamflow (Wood et al., 2002; Singla et al., 2012; Förster et al., 2018; Greuell et al., 2019), but also provide mechanistic explanations for the current and future state of hydrological systems.

In particular, the ensemble streamflow prediction (ESP; Day, 1985) technique has been used operationally by many forecasting

agencies in the world, and is considered a baseline for the implementation of dynamical forecasting frameworks (Wood et al., 2018). The approach relies on deterministic hydrologic model simulations forced with historical meteorological inputs up to the forecast initialization time, assuming that meteorological data and model are perfect, which yields initial hydrologic conditions (IHCs) without errors. Then, the model is forced with an ensemble of climate sequences, attributing all the streamflow forecast uncertainty to the spread of future meteorological forcings (FMFs). In the traditional ESP implementation,

each climate sequence (i.e., ensemble member) is drawn from a one-year observed meteorological time series, and the meteorological input traces associated with target years are excluded for hindcast generation/verification (Mendoza et al., 2017). Importantly, ESP cannot forecast extreme events with magnitudes that have not been recorded (Sabzipour et al., 2021), and forecast quality can be limited in non-stationary climates (Peñuela et al., 2020).

Because of its simplicity and relatively low cost, ESP has been widely used as a reference for developing and testing more

complex forecasting frameworks that incorporate dynamical climate model outputs to force hydrologic model simulations (e.g., Wood et al., 2005; Mo et al., 2012; Yuan et al., 2014; Arnal et al., 2018; Lucatero et al., 2018; Wanders et al., 2019; Peñuela et al., 2020; Baker et al., 2021). Notably, the approach remains a hard-to-beat benchmark when the target predictand is spring-summer snowmelt runoff (e.g., Arnal et al., 2018; Wanders et al., 2019), since it was originally designed to provide more skill for regions and times in the year where IHCs dominate the seasonal hydrologic response. This has motivated a large

body of research to improve ESP forecasts in snow-dominated areas, including evaluation and diagnostics of operational systems (e.g., Franz et al., 2003), the implementation of data assimilation methods (e.g., DeChant and Moradkhani, 2014; Huang et al., 2017; Micheletty et al., 2021), climate input selection (i.e., pre-ESP; Werner et al., 2004), statistical post-processing techniques (e.g., Wood and Schaake, 2008; Mendoza et al., 2017) and multi-model combination strategies (e.g., Bohn et al., 2010; Najafi and Moradkhani, 2015).

However, and despite the reliance of dynamical and some types of hybrid (i.e., statistical-dynamical; see review by Slater et al., 2022) approaches on hydrologic models, there has been limited attention on how parameter estimation strategies may affect



seasonal forecast quality. In particular, the choice of calibration metric is crucial because it involves defining the processes and/or target variables (including streamflow characteristics) that need to be well simulated for specific water resources applications (e.g., Pushpalatha et al., 2012; Pool et al., 2017; Mizukami et al., 2019).

In seasonal streamflow forecasting, the Nash-Sutcliffe efficiency (NSE; Nash and Sutcliffe, 1970) – a normalized version of the mean-square-error – is a common choice for single-objective (Yuan et al., 2013; Förster et al., 2018; Giuliani et al., 2020; Sabzipour et al., 2021) or multi-objective (e.g., Shi et al., 2008; Bohn et al., 2010) calibration frameworks. Other studies have preferred related metrics, like the mean-square-error (e.g., DeChant and Moradkhani, 2014), the root-mean-square error (e.g., Huang et al., 2017) and the mean absolute error (e.g., Yuan et al., 2013) between observed and simulated streamflow. Another

popular choice is the Kling-Gupta efficiency (KGE; Gupta et al., 2009), which has been applied to raw streamflow (e.g., Micheletty et al., 2021), root-squared flows (e.g., Crochemore et al., 2016; Harrigan et al., 2018) and inverse flows to emphasize low streamflow (Crochemore et al., 2017). The KGE has also been used in its non-parametric form (Pool et al., 2018) to capture different parts of the hydrograph (Donegan et al., 2021), or combined with NSE (e.g., Girons Lopez et al., 2021). Finally, seasonally-oriented metrics are attractive if the aim is to constrain the calibration process to the time window

of interest. For example, Yang et al. (2014) showed that calibrating hydrological model parameters using only data from the dry season improved forecast skill for months included therein in comparison to using the entire time series.

To the best of our knowledge, no previous studies have conducted a systematic assessment on how different types of calibration objective functions may impact forecast quality attributes and their relationship with catchment characteristics. Even more, it remains unclear whether 'good' seasonal forecasts are associated to calibration metrics that enable to reproduce the main

features of observed catchment behavior (i.e., hydrological consistency; Martinez and Gupta, 2010). This is a critical issue if hydrological models need to be operationally implemented for multiple purposes, since traditional objective functions may not necessarily reproduce streamflow characteristics described with different mathematical formulations (e.g., Mendoza et al., 2015). Therefore, we address the following research questions:

1. How dependent is the quality of seasonal streamflow forecasts on the choice of calibration metric and forecast
initialization times?

2. Is it possible to obtain skillful and reliable seasonal forecasts for the right reasons through an appropriate choice of calibration objective function?

3. How does the relationship between catchment characteristics and seasonal forecast quality vary for different calibration metrics?

To address these questions, we evaluate seasonal streamflow hindcasts produced with the ESP method, using three popular conceptual rainfall-runoff models calibrated with metrics that belong to different families of objective functions. We conduct our analyses over a collection of headwater basins in central Chile, where snow plays a key role in the hydrologic cycle (Masiokas et al., 2020; Mendoza et al., 2020; Murillo et al., 2022) and, especially, for streamflow predictability (Mendoza et al., 2014; Cornwell et al., 2016). Current operational practice in this region considers September-March (i.e., Spring and



Summer) water supply forecasts produced only once a year (September 1), based on subjectively adjusted outputs from statistical models that regress streamflow volumes against in situ measurements of precipitation, temperature, SWE, and antecedent streamflow, among other variables (DGA, 2022). Hence, this paper provides a baseline for ongoing and future streamflow forecasting efforts using dynamical and/or hybrid methods in central Chile. Additionally, the selected basins cover a wide range of physiographic and hydroclimatic characteristics (Alvarez-Garreton et al., 2018; Vásquez et al., 2021;

Sepúlveda et al., 2022), enabling the examination of possible connections between forecast quality and catchment attributes (Harrigan et al., 2018; Pechlivanidis et al., 2020; Donegan et al., 2021).

## 2 Study domain and data

We focus on 22 case study basins located in central Chile (28°-37°S, 70°-71°W), a domain that encompasses more than 60% of the country's population and, therefore, many socioeconomic activities that depend on water availability. The selected basins

are included in the CAMELS-CL dataset (Alvarez-Garreton et al., 2018) and meet the following criteria: (i) a low (i.e., < 0.05) human intervention index, which is defined as the ratio between annual volume of water assigned for permanent consumptive uses and the observed mean annual runoff; (ii) absence of large reservoirs; (iii) no major consumptive water withdrawals from the stream; (iv) seasonal snowmelt contributions to total runoff; (v) at least 75% of days with streamflow observations during the period April/1987 – March/2020; (vi) at least 20 water years with seasonal (Sep-Mar) streamflow observations for hindcast

verification purposes.

Figure 1 shows a suite of attributes for our case study basins, whose mean elevations and areas range between 1605 – 4275 m.a.s.l. and 81 – 4839 km2, respectively. The selected basins provide a pronounced hydroclimatic gradient, with aridity indices – defined as the ratio between mean annual potential evapotranspiration (PET) and mean annual precipitation (P) – spanning 0.5 – 7.0. Indeed, there is a north-south transition from semi-arid, water limited hydroclimates (with PET/P > 1) towards energy

limited environments (with PET/P < 1, see Figure 1c and 2d), with larger precipitation and runoff amounts. No clear spatial patterns are found in the fraction of precipitation falling as snow.

Figure 2 includes additional hydrological features for our sample of catchments. In terms of average seasonal patterns, higher Pardé coefficients are obtained in most basins during the snowmelt season (September-March, which spans the spring and summer seasons). Precipitation (Figure 2b) is concentrated between April and September, and intra-annual variations in PET

(Figure 2c) are consistent with seasonal temperature fluctuations in central Chile (not shown). Figure 2d also shows that the case study basins span different annual water and energy balances, complementing the latitudinal gradients shown in Figure 1). Aconcagua at Chacabuquito (ACO) is the only basin with a mean annual runoff ratio larger than 1, which can be explained by (i) underestimation of catchment-averaged precipitation, (ii) positive biases in runoff records, or (iii) glacier and/or groundwater contributions. Finally, the daily flow duration curves (FDCs; Figure 2e) show the diversity of hydrological

responses, with differences in high/low flows, mid-segment slope, median and other signatures.



We use daily time series of precipitation, mean air temperature, potential evapotranspiration (PET) and observed streamflow retrieved from the CAMELS-CL database (Alvarez-Garreton et al., 2018), which compiles information from different sources. Basin-averaged precipitation and mean temperature data are derived from the gridded observational product CR2MET (DGA, 2017; Boisier et al., 2018) version 2.0, which provides information of these variables for continental Chile at a 0.05° x 0.05°

horizontal resolution. PET is calculated with the formula proposed by Hargreaves and Samani (1985), and streamflow time series are acquired from stations maintained by the Chilean General Water Directorate (DGA), also available at the DGA's website (https://dga.mop.gob.cl/). Elevation data from the ASTER Global Digital Elevation Model (DEM), version 3.0 (U.S./Japan Aster Science Team), is used to generate hypsometric curves for the basins.

## 3    Approach

Figure 3 outlines our methodology, which includes four steps: (1) parameter calibration of three hydrological models (GR4J, TUW and SAC-SMA) configured in 22 snow-influenced basins using a suite of 12 objective functions; (2) seasonal (September-March) streamflow hindcast generation with the ESP method for 33 WYs (April/1987 - March/2020) and five initialization times, and evaluation of forecast quality attributes; (3) assessment of hydrological consistency through six streamflow signatures for the subset of best-performing objective functions in terms of forecast attributes, and (4) analysis of

possible relationships between catchment characteristics and ESP forecast attributes.

### 3.1    Hydrological modeling

#### 3.1.1    Models

We use three conceptual, bucket-style hydrological models: (i) GR4J (Perrin et al., 2003) coupled with the CemaNeige snow module (Valéry et al., 2014b); (ii) TUWmodel (Parajka et al., 2007), which follows the structure of HBV (Bergström, 1976);

and (iii) the Sacramento Soil Moisture Accounting (SAC-SMA; Burnash et al., 1973) model combined with SNOW-17 (Anderson, 1973) and a routing scheme (Lohmann et al., 1996). These model structures are widely used by the hydrology community (Addor and Melsen, 2019), with a myriad applications to streamflow forecasting. For example, SAC-SMA has been applied for testing alternative approaches (e.g., Mendoza et al., 2017), and is used to produce operational streamflow forecasts in the US (Micheletty et al., 2021). GR4J has been applied to assess streamflow forecasting frameworks in large

samples of catchments (e.g., Harrigan et al., 2018; Woldemeskel et al., 2018). HBV-like conceptual models have been used to assess short (e.g., Pauwels and De Lannoy, 2009; Verkade et al., 2013) to long (e.g., Peñuela et al., 2020) range streamflow forecasts, especially in European countries.

The GR4J model (Perrin et al., 2003) has a parsimonious structure consisting in two interconnected reservoirs and four free parameters. The CemaNeige module simulates snow accumulation and melt over five or more (user-defined; here we use 10)

elevation bands, using a two-parameter degree-day based scheme (Valéry et al., 2014b) that adds snowmelt to the soil moisture accounting reservoir. Water that is not intercepted or evaporated from the soil moisture accounting reservoir is partitioned into


two fluxes: one is routed with a unit hydrograph and then by a nonlinear routing store, and the other is routed using a single unit hydrograph.

The TUW model consists of four main routines. In the snow routine (with five free parameters), precipitation is partitioned
into snowfall and rainfall, and snow accumulation and melting are calculated with a degree-day scheme. Rainfall and snowmelt are inputs for the soil moisture routine (with three free parameters), which computes actual ET, soil moisture and runoff heading to the response area. The response routine (five free parameters) has an upper reservoir that produces surface runoff and interflow, and a lower reservoir producing baseflow. Finally, a routing scheme (two free parameters) delays total runoff using a triangular transfer function.

The SAC-SMA (Burnash et al., 1973) has a more complex structure than GR4J and TUW (with 16 free parameters), dividing the catchment into (1) an upper zone that simulates hydrological processes occurring in the root, surface, and atmospheric zones, producing surface and direct runoff; and (2) a lower zone, where percolation occurs and baseflow is produced. The model is coupled with the conceptual snow accumulation and ablation model SNOW-17 (Anderson, 1973), which simulates snow accumulation and melt using a simplified energy balance and requires the specification of 10 free parameters. An
independent, two-parameter routing scheme, based on the linearized Saint-Venant equation, is used to route runoff and baseflow (Lohmann et al., 1996).

Here, we use model versions from open-source packages implemented in the statistical software "R" (http://www.r-project.org/). GR4J and CemaNeige (hereafter referred to as GR4J) are implemented in the open-source package "*airGR*" (Coron et al., 2017), whereas TUW and SAC are available in the packages "*TUWmodel*" (Viglione and Parajka, 2020) and
"*sacsmaR*" (Taner, 2019), respectively. All the models require daily time series of catchment-scale precipitation (P, mm), PET (mm) and mean air temperature (T, °C), and are implemented in a lumped fashion. We stress that the use of three models does not seek to provide comparisons among different model structures; instead, we aim to examine to what degree our results and conclusions can be model-dependent.

### 3.1.2 Calibration strategy

We calibrate model parameters (Figure 3a) using the global optimization algorithm Shuffled Complex Evolution (SCE-UA; Duan et al., 1992), implemented in the R package "*rtop*" (Skøien et al., 2014). We use the period April/1986 – March/1994 for model spin-up and compute the calibration objective function using modeled and observed streamflow data from the period April/1994 – March/2013 because it spans a diverse range of hydroclimatic conditions. For each model and basin, we perform 12 calibrations using the objective functions listed in Table 1. Eight metrics (groups 1-4) are selected because they are
representative of different families of objective functions and have been widely used for various modeling purposes. For example, the NSE with flows in log space (Log-NSE) has been used to enhance low flow simulations (e.g., Oudin et al., 2008; Melsen et al., 2019), while the recently proposed Split KGE (Fowler et al., 2018) aims to provide robust streamflow simulations under contrasting climatic conditions. Additionally, we include four calibration metrics formulated to improve seasonal streamflow simulations. The equations for each objective function are provided in Table S1 (Supporting Information). Model





evaluation is performed in two periods: (i) April/1987 - March/1994, which is hydroclimatically diverse, and (ii) April/2013 – March/2020, which is characterized by unprecedented and temporally persistent dry conditions (Garreaud et al., 2017, 2019). In both cases, the preceding 8-year period is used for model spin-up.

## 3.2 Hindcast generation and evaluation

We produce seasonal streamflow forecasts by retrospectively applying the ESP method for the period April/1987 – March/2020

(Figure 3b), using five initialization times (from May 1 to September 1). Hence, for each combination of catchment, hydrological model, parameter set (i.e., objective function) and initialization time, we complete the following steps:

1. Force model simulations during the eight water years preceding the initialization time $t_i$ to obtain the initial hydrologic conditions (IHCs). This is similar to the approach adopted by Shi et al. (2008), who initialized the model with a 10-yr spin-up simulation prior to each of the forecast dates.

2. Using the states obtained in step 1, run hydrologic model simulations using observed meteorological data from the remaining 32 water years (i.e., the forcings of the year to be forecasted are not used), generating an ensemble of 32 traces for year $n$.

3. Aggregate daily streamflow volumes within the period of interest (September 1 – March 31), obtaining an ensemble of 32 seasonal streamflow forecasts.

Steps 1-3 are repeated until a time series of 33 ensemble seasonal streamflow hindcasts is obtained. Then, we evaluate different forecast quality attributes using a set of deterministic and probabilistic metrics (Table 2). These include standard measures such as the coefficient of determination ($R^2$), the percent bias, and the root mean squared error (*RMSE*). All deterministic metrics are calculated using the ensemble median. Probabilistic skill is assessed through the continuous ranked probability score (CRPS; Hersbach, 2000), which measures the temporal average error between the forecast cumulative distribution function (CDF) and that from the observation. We compute the continuous ranked probability skill score (CRPSS) using the

observed mean climatology as the reference forecast, instead of modeled data as in other studies (e.g., Harrigan et al., 2018; Crochemore et al., 2020), making our verification results independent from the choice of objective function and hydrological model. Forecast reliability – i.e., adequacy of the forecast ensemble spread to represent the uncertainty in observations – is evaluated using the α index from the predictive quantile-quantile (QQ) plot (Renard et al., 2010). QQ plots compare the empirical CDF of forecast $p$-values (i.e. $P_i(o_i)$, where $P_i$ and $o_i$ are the forecast CDF and observation at year $i$) with that from

a uniform distribution $U[0,1]$ (Laio and Tamea, 2007).

## 3.3 Assessment of hydrological consistency

From each family of objective functions listed in Table 1, we choose the one providing the overall best hindcast performance (quantified through the median from the sample of catchments) for all combinations of initialization time, performance metric

and model and evaluate its capability to provide hydrologically consistent simulations (Figure 3c) using six signature metrics





of hydrological behavior. Our goal here is to explore the extent to which the quality of seasonal streamflow forecasts – achieved with a specific calibration objective function – is connected to the model's capability to reproduce streamflow characteristics. Hence, we select metrics that cover various aspects of simulated catchment response, including precipitation partitioning into ET and runoff, high and low flow volumes, frequency of high flow events, flashiness of runoff and medium flows. The notation,

short description, mathematical formulation, and physical process associated with each evaluation metric are detailed in Table 3.

We also examine possible variations (gain/loss) in a specific forecast performance measure when selecting popular (i.e., NSE and KGE) or alternative calibration metrics that yield hydrologically consistent simulations ($M_{OF}$), relative to reference forecasts obtained with the overall best objective function in terms of hindcast performance ($M_{REF}$). For example, the variation

in CRPSS is obtained as:

$$\Delta CRPSS = CRPSS_{OF} - CRPSS_{REF} \qquad\qquad\qquad (1)$$

It should be noted that Eq. (1) can be applied to any metric of Table 2. Here, we use it for forecasts initialized on September 1, evaluating changes in CRPSS, the $\alpha$ reliability index and the bias in flow volumes.

### 3.4 Drivers of seasonal streamflow predictability

To explore possible relationships between the quality of seasonal streamflow forecasts and catchment characteristics, we compute, for each combination of hydrological model, initialization time and objective function, the Spearman's rank correlation coefficient between forecast performance measures – namely, the CRPSS, the $\alpha$ reliability index, and the coefficient of determination $R^2$ – and selected physiographic-hydroclimatic descriptors (Figure 2d). To this end, we use the five calibration metrics from section 3.3 and the basin descriptors in Table 4. All the forecast evaluation metrics are calculated using the entire

time series (i.e., 33 WYs).

## 4 Results

### 4.1 Example: ESP results at the Upper Maipo River basin

Figure 4 illustrates the hindcasting framework used in this study for the Maipo at El Manzano basin (4,839 km$^2$), which provides nearly 70% of municipal water supply for Santiago (Chile's capital city) and is also the primary source of water for

agriculture, hydropower, and industry in the area (Ayala et al., 2020). Specifically, Figure 4 shows sample results of seasonal (i.e., September - March) streamflow forecasts initialized on July 1 and September 1 for the period April/1987 – March/2020, using parameter sets obtained with three calibration objective functions and the three hydrological models. As expected, the forecast initialization time greatly impacts CRPSS and $R^2$ regardless of calibration metric and model, with substantial improvements towards the beginning of the snowmelt season; conversely, the $\alpha$ reliability index decreases as we approach

September 1 (the forecast ensemble becomes narrower). The results also show that, for those initialization times where IHCs





(in particular, snow accumulation for this domain) play a key role on streamflow predictability, the choice of calibration criteria may have large effects on CRPSS (e.g., September 1 for GR4J) and α (e.g., September 1 for TUW), in contrast to forecasts initialized on July 1 or earlier dates (not shown). Further, VE-Sep yields the highest performance measures when GR4J and TUW models are used.

**4.2 Effects on hindcast performance**

Figure 5 and Figure 6 show CRPSS and percent bias results, respectively, for our sample of catchments and all initialization times, using the three hydrological models and parameter values obtained with different calibration objective functions. In general, the seasonal objective functions (yellow boxplots) provide the highest median values across basins for 57 out of 75 combinations (3 models x 5 performance metrics x 5 initialization times). The highest median performance metric with the

TUW model is mainly obtained through seasonal objective functions (11 out of 25 cases, with VE-Sep standing out) and KGE-based metrics (11 out of 25 cases, with ModKGE standing out). When using the GR4J and SAC models, seasonal objective functions dominate, being VE-Sep and KGEV-Sep the best-performing in most cases, respectively. On the other hand, KGE(Q)+KGE(1/Q) and Split KGE generally yield the poorest forecast quality across hydrological models. Interestingly, some objective functions enhance the spread in performance metrics across basins – e.g., see CRPSS values obtained with

GR4J and SAC; α indices (Figure S1) and RMSE (Figure S3) obtained with SAC using KGE(Q)+KGE(1/Q) as calibration metric.

The catchment sample means of all forecast verification metrics (Table 2) obtained from objective functions belonging to the same family are not significantly different (p-values > 0.05 from t-tests, not shown), which is valid for the different initialization times evaluated. However, there are significant differences between verification means obtained with the best

and the worst performing calibration metrics. For example, see CRPSS results for September 1 forecasts obtained from the TUW model (Figure 5), calibrated with VE-Oct versus Split KGE (p-value = 0.03). For hindcasts initialized before July 1, when the signal from IHCs is weak, the choice of calibration metric becomes less relevant, and the magnitude of differences depends on the forecast verification criteria. For example, significant differences in percent bias are obtained between seasonal and meta-objective seasonal functions, though this is not the case for CRPSS and the α index. Based on these results and

additional analyses with the α index, $R^2$ and NRMSE (Figures S1-S3), we select the overall best-performing (or "representative") objective function from each family (Table 1) for further analyses, namely NSE, ModKGE, Split KGE, VE-Sep and KGE(Q)+NSE(log(Q)).

Figure 7 illustrates how initialization time affects forecast quality attributes when using the five representative calibration metrics. As shown for the Upper Maipo River basin (Figure 4), CRPSS and $R^2$ (the α index) improve (degrades) as forecast

initializations approach September 1, with considerable increments in skill on July 1 compared to May 1 and June 1 hindcasts. Indeed, the skill of May 1 forecasts is rather low (CRPSS ranging 0.24-0.27, 0.26-0.28, and 0.25-0.27 for GR4J, TUW, and SAC models, respectively) and does not improve significantly on June 1. Additionally, for a given model and calibration



objective function, inter-basin differences in CRPSS increase as forecast initializations approach the beginning of the snowmelt season.

An interesting feature of Figure 7 is that, in general, the TUW model outperforms the GR4J and SAC models in terms of CRPSS and the α index across all initialization times and objective functions.

### 4.3   Seasonal forecast quality vs. hydrological consistency

We now turn our attention to the following question: to what extent is the quality of seasonal streamflow forecasts related to the proper simulation of streamflow characteristics? Figure 8 shows daily hydrographs and seasonal variation curves for the

Maipo at El Manzano River basin, obtained with the five representative objective functions and three hydrological models. Although these calibration metrics yield the best forecast quality attributes within each family, the simulated hydrographs can be very different, particularly during the target forecast period (September-March). Specifically, VE-Sep (Figure a.5) yields parameter values that cannot properly reproduce daily streamflow dynamics (KGE = 0.45-0.53), while the other objective functions provide a more realistic representation of runoff seasonality during validation periods 1 and 2 (Panels b.1-c.4). The

split KGE stands out with a higher $R^2$ (> 0.85) using the TUW and SAC models (Figure 8c.3) during the validation period 2 (drought period) compared to the other calibration metrics, a result that is replicated in the remaining basins (not shown). Figure 9 displays biases in hydrological signatures obtained with the five selected calibration metrics for all basins (shown in boxplots). Overall, the results show that there is no unique best objective function for the signatures examined here. For instance, if the selected model is TUW, Split KGE and KGE(Q)+NSE(log(Q)) provide the lowest biases for FHV (FLV) during

the three time periods, while NSE and Split KGE yield the lowest biases in mid-range flows (FMM) during calibration and validation 1. Additionally, the calibration metric providing the lowest mean absolute bias (from all signatures and basins) may vary depending on the selected model and evaluation period. For example, ModKGE (KGE(Q)+NSE(log(Q))) yields the lowest mean absolute bias for TUW and SAC (GR4J) during calibration, while NSE, KGE(Q)+NSE(log(Q)) and Split KGE provide the lowest mean absolute bias during validation 2 for TUW, GR4J and SAC, respectively. As expected, VE-Sep yields

the poorest hydrological consistency across all periods and models compared to the other calibration metrics. Interestingly, some objective functions enhance inter-basin differences in signature biases (e.g., compare the spread in RR bias obtained with KGE(Q)+NSE(log(Q)) and Split KGE for the calibration period), and 'well-behaved' OFs in terms of mean absolute signature bias may produce significantly different mean biases for specific signatures (e.g., FLV, FMM and FMS obtained with Split KGE or ModKGE for specific models/periods).

What would be the impacts of selecting a calibration metric that yields good hydrological consistency, instead of a reference objective function that provides the overall best forecast performance? Figure 10 displays variations in CRPSS, α indices and biases (obtained with equation 1) using VE-Sep (which also yields the largest biases in hydrological signatures) as the reference, for forecasts initialized on September 1. One can note that Split KGE (one of the OFs with the highest hydrological consistency) yields worse forecast performance than the reference (ΔCRPSS and Δα ~ -0.1, and a 17% average increase in

percent bias for the three models), with similar results for KGE(Q)+KGE(1/Q). Figure 10 also shows that popular performance





metrics (i.e., NSE and KGE) yield overall similar results in terms of hydrological consistency and forecast performance. Interestingly, KGE(Q)+NSE(log(Q)) offers a good compromise between these two features, with the smallest loss in forecast quality ($\Delta$CRPSS ~ -0.03, $\Delta\alpha$ ~ -0.06 and 8% increase in bias) among the metrics examined in Figure 10.

### 4.4 Forecast quality vs. catchment characteristics

Figure 11 explores the factors that control seasonal forecast quality, and the extent to which the choice of calibration metric impacts the connections inferred from our sample of catchments. We find statistically significant correlations between CRPSS and the baseflow index ($\rho$ ~ 0.2 – 0.8), being VE-Sep ($\rho = 0.70$), ModKGE ($\rho = 0.49$), and VE-Sep ($\rho = 0.41$) the objective functions that maximize such relationship for September 1 when using GR4J, TUW, and SAC, respectively. Figure 11 shows significant correlations between CRPSS and the interannual variability of runoff ($\rho$ ~ 0.0 – 0.6) – especially for September 1

forecasts ($\rho = 0.64$ for ModKGE/GR4J, $\rho = 0.53$ for VE-Sep/TUW, and $\rho = 0.62$ for VE-Sep/SAC). Also positive, but generally weaker correlations are obtained between forecast skill and p-seasonality ($\rho$ ~ -0.6 – 0.0), as well as the fraction of precipitation falling as snow ($\rho$ ~ 0.0 – 0.4).

Overall, the $\alpha$ reliability index (Figure 11, center panels) correlates differently than CRPSS with basin characteristics, with generally smaller values that range between -0.4 and 0.4. Although negative correlations are obtained between interannual

runoff variability and $\alpha$ for all models, largest and significant absolute values are obtained for September 1 forecasts only with the GR4J and SAC models. The right panels in Figure 11 show that some catchment descriptors (e.g., baseflow index, interannual variability in runoff) yield similar correlations with $R^2$ compared to those obtained with CRPSS.

In general, the choice of calibration metric affects more the strength of the relationships between forecast quality and catchment attributes, rather than the sign, regardless of the model used. In particular we find that the correlations between CRPSS and

catchment descriptors obtained with Split KGE (which maximizes hydrologic consistency), are weaker than those obtained with other calibration metrics (e.g., see results for baseflow index with TUW, interannual runoff variability with all models, and fraction of precipitation falling as snow with all models).

## 5 Discussion

### 5.1 Compromise between hydrological consistency and forecast performance

The experiments presented here provide important insights on the impacts that calibration metric selection may have on the performance of dynamical seasonal forecasting systems in snow-influenced environments, in particular for the traditional ESP technique. Despite the choice of calibration metric is a big topic in the hydrologic modeling literature, given the implications for a myriad of water resources applications (see, for example, Shafii and Tolson, 2015; Pool et al., 2017; Melsen et al., 2019; Mizukami et al., 2019), it has received limited attention for the specific case of ensemble seasonal forecasting. Additionally,



our sample of catchments offers an interesting experimental setup, spanning an ample range of mountain hydroclimates and physiographic characteristics.

The results presented here reveal tradeoffs between forecasting skill and hydrological consistency in model simulations. Despite seasonal OFs produced the best forecast performance regardless of the hydrological model, they did not result in acceptable hydrological consistency, which was better achieved with time-based meta-objective functions (Split-KGE) or through meta-objective functions with transforms (KGE(Q)+KGE(1/Q) and KGE(Q)+NSE(log(Q))). Conversely, these objective functions produced worse forecast performance than the reference calibration metric (e.g., a 16% and 14% loss in CRPSS for forecasts initialized on September 1 using Split KGE and KGE(Q)+NSE(log(Q)), respectively). These results highlight the risk of selecting model configurations for a specific purpose without complementary insights on the representation of features that may be useful for other operational applications. Among the options examined here, KGE(Q)+NSE(log(Q)) provided the best compromise between hydrological consistency and forecast performance, with only a 4% loss in CRPSS for September 1 forecasts.

## 5.2 Initialization times and forecast skill

ESP forecasts produced at the beginning of the snowmelt season for our set of catchments are very skillful (CRPSS ~ 0.62-0.67 for seasonal OF, CRPSS ~ 0.60-0.64 for meta-objective with transformations OF, and 0.60-0.62 for KGE-type OF), and the skill decreased monotonically with longer lead times, regardless of the choice of calibration OF and model. For example, the average CRPSS of June 1 forecasts is 0.31 (0.29) for seasonal (meta-objective) OFs. Importantly, forecast skill improves considerably between June 1 and July 1, reflecting that the information on snow accumulation collected at the end of fall and beginning of the winter season is crucial to maximize the predictability from IHCs in Andean catchments. These results align well with previous studies in other snow-influenced mountain environments and cold regions of the world, such as the Colorado River basin (Franz et al., 2003; Baker et al., 2021), the US Pacific Northwest (Mendoza et al., 2017) and Northern Europe (Pechlivanidis et al., 2020; Girons Lopez et al., 2021). More generally, this study reinforces – through multiple hydrologic model setups – the decay of ESP forecast skill with lead time, which has been also reported in domains where snow has a limited influence on the water cycle (e.g., Harrigan et al., 2018; Donegan et al., 2021).

## 5.3 Factors controlling seasonal forecast quality

Our results reaffirm that seasonal forecast quality is better in slow-reacting basins with a higher baseflow contribution (Harrigan et al., 2018; Pechlivanidis et al., 2020; Donegan et al., 2021; Girons Lopez et al., 2021), and with a higher amount of precipitation falling as snow (i.e., snow-dominated river systems), in agreement with previous studies conducted over large domains (e.g., Arnal et al., 2018; Wanders et al., 2019). In our study area, seasonal forecast quality is also explained by high interannual runoff variability – with significant correlations in September 1 and August 1 –, which is a characteristic feature of snow-dominated catchments in Central Chile (i.e., north from 37°S), where year to year variability in mean annual precipitation is also considerable (Hernandez et al., 2022). In the driest (northernmost) catchments, only a few sporadic storms



contribute to annual precipitation amounts (Hernandez et al., 2022), and the high skewness of daily runoff challenges the calibration of hydrological models. On the other hand, the predictability from future meteorological forcings becomes important in the wetter southern hydroclimates, since occasional spring precipitation events may have a strong effect on total

spring-summer runoff volumes.

## 5.4 Inter-model differences

In this study, we obtained similar effects of calibration criteria selection across model structures, though these provide differences in forecast performance and hydrological consistency. Despite the three models are in the lower zone of the spatial–process complexity continuum (Hrachowitz and Clark, 2017), they greatly differ in the number of parameters, which does not

necessarily relate to seasonal forecast quality. In fact, the TUW model (15 parameters) provides generally better ESP forecasts than GR4J (6 parameters) and SAC-SMA (28 parameters). In addition to discrepancies related to soil storages and associated parameterizations, the models differ in terms of their snow modules – which is a key component for seasonal predictability in mountainous basins – , with 2, 5 and 10 free-parameters within GR4J, TUW and SAC-SMA, respectively. The snow routines used in GR4J (CemaNeige; Valéry et al., 2014b) and TUW (Parajka et al., 2007) models follow a simple degree-day factor

approach, differing mainly in the characterization of precipitation phase (TUW allows for a mix of rain and snow) and the melt temperature threshold (set as 0°C for GR4J and defined as a free-parameter in TUW). On the other hand, Snow-17 (snow routine coupled to SAC-SMA) is based on a simplified energy balance (Anderson, 1973). Both CemaNeige and Snow-17 models estimate precipitation phase using a single temperature threshold (i.e., precipitation can occur only as rain or snow). Finally, both TUW snow routine and the Snow-17 model include a parameter to correct snowfall undercatch.

The results presented here, the inter-model differences described above and previous work on the implications of precipitation phase partitioning (Harder and Pomeroy, 2014; e.g., Valéry et al., 2014a; Harpold et al., 2017) suggest that a gradual transition between rain and snow (as in the TUW model) may favor seasonal streamflow forecast performance in snow-influenced regimes, especially in catchments with large elevation ranges and extended snowmelt seasons (Girons Lopez et al., 2020). However, testing such hypothesis is out of the scope of this study, for which controlled modeling experiments would be

required.

## 5.5 Limitations and future work

In this study, we used a global, single-objective optimization algorithm to find the "best" parameter set given a combination of forcing, model structure and calibration objective function; hence, we did not explore the potential effects of parameter equifinality on the partitioning of internal fluxes (flux equifinality; Khatami et al., 2019) – which may affect the simulation of

hydrological signatures (e.g., baseflow index, flashiness of runoff) – or the quality of seasonal forecasts. Further, our assessment of hydrological consistency is solely based on the model's ability to reproduce streamflow characteristics, though snow depth (Tuo et al., 2018; Sleziak et al., 2020), snow water equivalent (e.g., Nemri and Kinnard, 2020), snow covered area (e.g., Şorman et al., 2009; Duethmann et al., 2014), or the combination of these and other in-situ or remotely sensed variables





(e.g., Kunnath-Poovakka et al., 2016; Nijzink et al., 2018; Tong et al., 2020) could be incorporated to achieve a more exhaustive
evaluation of model realism. Moreover, multivariate calibration methods using multi-objective optimization algorithms (e.g.,
Yapo et al., 1998; Pokhrel et al., 2012; Shafii and Tolson, 2015) may be considered to examine potential improvements in
hydrological consistency and streamflow forecast quality compared to traditional parameter estimation approaches.

The data, models and results obtained here provide a test bed for the systematic implementation of new tools aimed at
improving seasonal streamflow forecasts across central Chile. Ongoing work is focused on developing a historical ensemble
gridded meteorological product for this study area, the implementation of data assimilation methods for improved estimates
of initial conditions, the assessment of seasonal climate forecast products and the inclusion of additional catchments. Given
the strong relationships between basin-scale hydrology in this domain and some large-scale climate patterns (e.g., El Niño
Southern Oscillation; Hernandez et al., 2022), future research should explore the potential of post-processing techniques that
take advantage of climate information to improve forecast quality (e.g., Hamlet and Lettenmaier, 1999; Werner et al., 2004;
Yuan and Zhu, 2018; Donegan et al., 2021). Finally, the forecast generation and evaluation analyses presented here should be
extended to fall and winter seasons, which are relevant for domestic water supply and other applications.

## 6  Conclusions

Dynamical systems have been implemented by many organizations across the globe for operational seasonal streamflow
forecasting. Despite their reliance on hydrological models, no detailed assessments have been conducted to understand how
the choice of calibration metric affects the quality attributes of seasonal streamflow forecasts, their connection with simulated
streamflow characteristics and the relationship between forecast quality and catchment descriptors. Here, we provide important
insights using the traditional ensemble streamflow prediction (ESP) method to generate seasonal hindcasts of spring/summer
streamflow in 22 basins in central Chile, where snow plays a key role in the hydrologic cycle. We use three popular conceptual
rainfall-runoff models calibrated with 12 metrics from different families of objective functions. The main conclusions are:


- The choice of calibration metric yields significant differences in forecast quality (except $R^2$) for hindcasts initialized
  during winter months. Such effect decreases considerably for hindcasts initialized during the fall season.
- The comparison of seasonal forecasts obtained from different families of objective functions revealed that
  hydrological consistency does not ensure satisfactory seasonal ESP forecasts (e.g., Split KGE), and that satisfactory
ESP forecasts are not necessarily associated to a hydrologically consistent parameter set (e.g., VE-Sep).
- We could identify at least one objective function (KGE(Q)+NSE(log(Q))) that yields a reasonable balance between
  hydrological consistency and forecast performance.
- The baseflow index and the interannual runoff variability are the strongest predictors of probabilistic skill and $R^2$
  across objective functions and models. Moreover, the choice of calibration metric generally affects the strength of the
relationship between forecast quality and catchment attributes.



The results presented here highlight the importance of hydrologic model calibration in producing skillful seasonal streamflow forecasts and drawing robust conclusions on hydrological predictability. Improving parameter estimation strategies can benefit not only operational systems relying on dynamical methods but also a myriad of hybrid approaches designed to leverage information from hydrologic model outputs. By advancing our understanding of the complex interplay between calibration metrics, model performance, and catchment characteristics, our study contributes to the ongoing effort to enhance the accuracy and reliability of streamflow forecasts in snow-influenced domains, in order to support informed water resources management decisions.

## 7    Code availability

All the data and models used to produce the results included in this paper here are publicly available at Zenodo (Araya et al., 2023; https://doi.org/10.5281/zenodo.7853556).

## 8    Author contributions

DA, PM and EMC conceptualized the study and designed the overall approach. DA conducted all the model simulations, generated the hindcasts, analyzed the results and created all the figures. PM and EMC provided support to set up the scripts used in this study. All the authors contributed to refine the methodology and analysis framework, discussed the results and contributed to writing, reviewing and editing the manuscript.

## 9    Competing interests

The authors declare that they have no conflict of interest.

## 10    Acknowledgments

Pablo A. Mendoza received support from Fondecyt Project 11200142. J. McPhee and P.A. Mendoza received support from CONICYT/PIA Project AFB180004.

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



**Table 1. Objective functions used for model calibration. The bold text indicates the notation used in this paper.**

| Group of objective function | Objective functions utilized | Description | Reason for use and attributes |
|---|---|---|---|
| 1. Classic least squares | **NSE** (Nash and Sutcliffe, 1970). | Normalized variant of the Mean Square Error (MSE). It minimizes the ratio of the variance of the simulated flows to the variance of the observed flows. | One of the most widely used metrics to assess the predictive skill of hydrological models. |
| 2. Least squares variations | **KGE** (Gupta et al., 2009); **KGE'** (Kling et al., 2012); **ModKGE** (Mizukami et al., 2019); **KGE''** (Tang et al., 2021) | Focus on optimizing three aspects of the time series: variability, bias, and correlation. | Popular family of metrics that combine the NSE components (i.e., correlation, bias, variability) in a more balanced fashion. |
| 3. Time-based meta-objective functions | **Split KGE** (Fowler et al., 2018). The KGE (Gupta et al., 2009) is calculated separately for each year, and the annual values are averaged. | Consider different sub-periods of the calibration period, in which a value of the metric is calculated and then combined into a single meta-objective function (e.g., average). | Reducing the year-to-year variability of model performance would allow for a stable set of parameters over time. Each subperiod has the same weight in the calculation of the metric. |
| 4. Meta-objective functions with transforms | **KGE(Q)+KGE(1/Q)** **KGE(Q)+NSE(log(Q))** | Linear combination of performance metrics that may consider transformations (e.g., using the inverse of the runoff or the logarithm). | The transformations emphasize medium and low flows. The weighting allows to consider high and low flows simultaneously. |
| 5. Seasonal objective functions | Seasonal (Sep-Mar) RMSE (**VE-Sep**); Seasonal (Oct-Mar) RMSE (**VE-Oct**); Seasonal (Sep-Mar) KGE (**KGEV-Sep**); Seasonal (Oct-Mar) KGE (**KGEV-Oct**). | The daily values are aggregated (i.e., summed) to generate a yearly time series with seasonal runoff volumes. Then, the sum of squares is minimized for all the time steps (i.e., WYs) within the calibration period. | Since the predictand is seasonal volume, testing metrics that focus on optimizing volume seems logical. However, this approach has the disadvantage of misrepresenting streamflow dynamics at finer time scales (e.g., daily or monthly). |






**Table 2. Performance metrics used to evaluate seasonal streamflow forecasts.**

| Name | Equation | Description |
|---|---|---|
| Coefficient of determination | $R^2 = \left( \dfrac{\sum_{i=1}^{N}(q_{m,i}-\overline{q_m})(o_i-\overline{o})}{\sqrt{\sum_{i=1}^{N}(q_{m,i}-\overline{q_m})^2} \cdot \sqrt{\sum_{i=1}^{N}(o_i-\overline{o})^2}} \right)^2$ | Deterministic metric that varies $[0,1]$ with a perfect score of 1. It measures the linear association between forecasts and observations. |
| Percent bias | $\%bias = \dfrac{\sum_{i=1}^{N}(q_{m,i}-o_i)}{\sum_{i=1}^{N} o_i} \cdot 100$ | Deterministic metric that varies $(-\infty, \infty)$, with perfect score of 0. It measures the difference between the mean of the forecasts and the mean of observations. |
| Normalized root mean squared error | $NRMSE = \dfrac{\sqrt{\dfrac{1}{N}\sum_{i=1}^{N}(q_{m,i}-o_i)}}{sd(o_i)} \cdot 100$ | Deterministic metric that varies $[0, \infty)$, with perfect score of 0. |
| Continuous ranked probability skill score | $CRPSS = 1 - \dfrac{\overline{CRPS_{fcst}}}{\overline{CRPS_{ref}}}$ $CRPS = \dfrac{1}{N}\sum_{i=1}^{N}\int_{-\infty}^{\infty}[F(q)-F_0(q)]^2\,dq$ $F_0(q) = \begin{cases} 0, & q < 0 \\ 1, & q > 0 \end{cases}$ | Probabilistic metric that varies $(-\infty, 1]$, with perfect score of 1. It measures the skill of CRPS relative to a reference forecast (Hersbach, 2000). CRPS quantifies the difference between the CDF of a forecast ($F$), and the corresponding CDF of the observations ($F_o$). |
| $\alpha$ reliability index | $\alpha = 1 - 2\left[ \dfrac{1}{N}\sum_{i=1}^{N}\lvert P_i(o_i) - U(o_i) \rvert \right]$ | Probabilistic metric that varies $[0, 1]$. It quantifies the closeness between the empirical CDF of sample p-values with the CDF of a uniform distribution. A value of 0 is the worst, and 1 reflects perfect reliability (Renard et al., 2010). |

$q_{m,i}$: Forecast ensemble median for year $i$
$\overline{q_{m,i}}$: Average over forecast ensemble medians
$o_i$: Observation for year $i$
$\overline{o_i}$: Average of observations
$P_i(o_i)$: Non-exceedance probability of $o_i$ using ensemble forecast for year $i$
$U(o_i)$: Non-exceedance probability of $o_i$ using the uniform distribution U $[0,1]$




**Table 3. Hydrological signatures used to evaluate the models' capability to generate hydrologically consistent simulations.**

| Notation | Short description | Equation | Hydrologic process |
|---|---|---|---|
| RR | Runoff ratio | $RR = \overline{Q}/\overline{P}$ | Overall water balance. |
| F9M | N° of days with flows greater than nine times the median | $\lvert F9M \rvert = \left\lvert \{q_i \in Q \mid q_i > 9 \cdot \tilde{Q}\} \right\rvert$ | Measure of the capacity of the catchment to represent extreme events. |
| FHV | FDC high-segment volume | $FHV = \sum_{h}^{H} q_h$ | Measure of the catchment reaction to large rainfall/snowmelt events. |
| FLV | FDC low-segment volume | $FLV = \sum_{l}^{L} [\log(q_l) - \log(q_L)]$ | Measure of the long-term baseflow processes. |
| FMS | FDC mid-segment slope | $FMS = \dfrac{\log(q_m) - \log(q_M)}{m - M}$ | Measure of the catchment reactivity or flashiness. |
| FMM | FDC median | $FMM = \tilde{Q}$ | Measure of mid-range flows. |

$\overline{Q}$: Average of a basin's runoff time series $(Q)$
$\overline{P}$: Average of a basin's precipitation time series $(P)$
$\tilde{Q}$: Runoff median value
$q_i$: Runoff observation/simulation for day $i$
$q_h$: Runoff observation/simulation for flows with exceedance probabilities lower than 0.02 in the FDC
$q_l$: Runoff observation/simulation for flows with exceedance probabilities greater than 0.70 in the FDC
$q_L$: Minimum runoff observation/simulation
$q_m$: Runoff observation/simulation with exceedance probability of 0.20
$q_M$: Runoff observation/simulation with exceedance probability of 0.70






**Table 4. Selected physiographic and climatic characteristics to explore drivers of seasonal forecast quality. Hydroclimatic attributes are computed for the period April/1987 – March/2020.**

| Name | Description | Units | Data source | Reference |
|---|---|---|---|---|
| Aridity index (AI) | Aridity calculated as the ratio of mean annual PET to mean annual precipitation | - | Computed for the study period | Budyko (1974) |
| Fraction of precipitation falling as snow | Fraction calculated as a function of temperature and a variable that quantifies the seasonal variation of precipitation, and its temporal distribution | - | CAMELS-CL dataset | Eq. (13) in Woods (2009) |
| p-seasonality | Seasonality of precipitation. Positive (negative) values indicate that precipitation peaks occur in summer (winter); values close to 0 indicate uniform precipitation all over the year | - | CAMELS-CL dataset | Eq. (14) in Woods (2009) |
| Interannual runoff variability | Coefficient of variation for the time series of annual runoff | - | Computed for the study period | - |
| Baseflow index | Computed as ratio of mean daily baseflow to mean daily discharge | - | CAMELS-CL dataset | Ladson et al. (2013) |
| Mean elevation | Catchment mean elevation | m.a.s.l. | CAMELS-CL dataset | ASTER GDEM, Tachikawa et al. (2011) |
| Fraction of the basin covered by forest | Fraction of the catchment covered by forest according to a land cover map. Includes native forest and forest plantation | - | CAMELS-CL dataset | Zhao et al. (2016) |
| Fraction of the basin covered by barren land | Fraction of the catchment covered by barren land according to a land cover map. Includes dry salt flats, sandy areas, and bare exposed rocks | - | CAMELS-CL dataset | Zhao et al. (2016) |






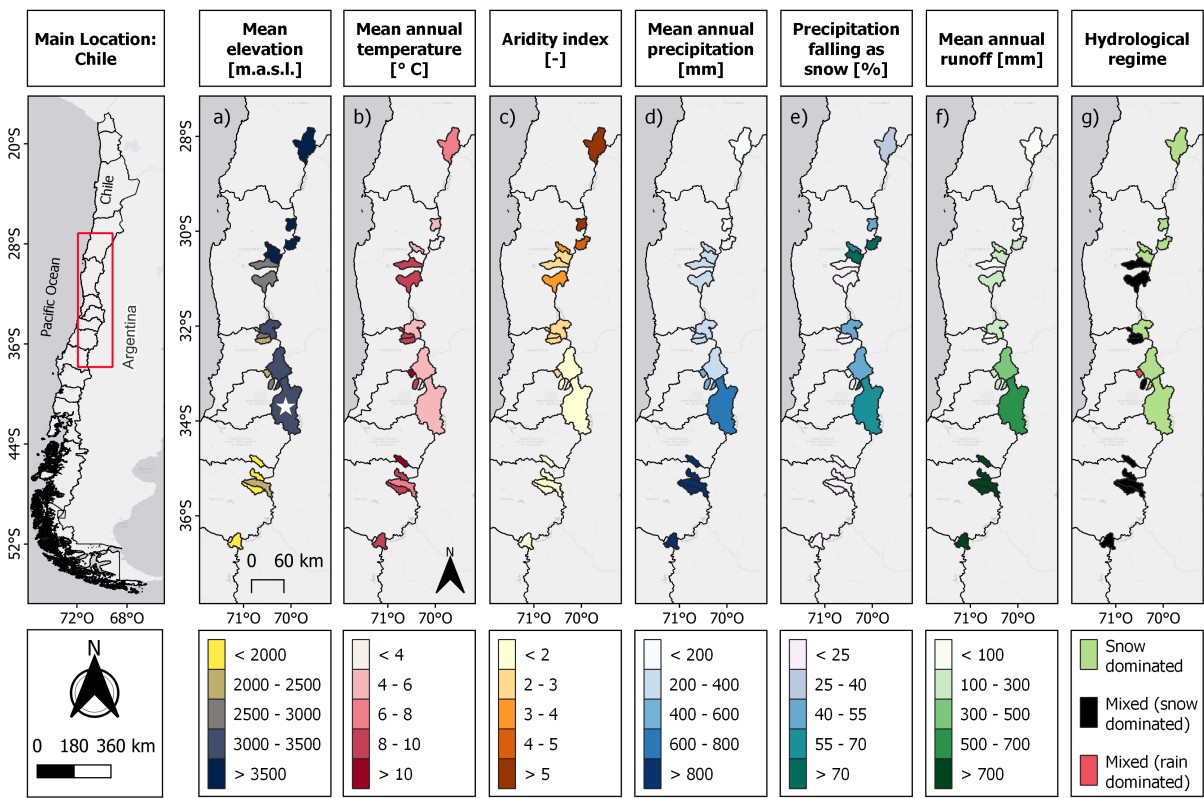

Figure 1. Location and spatial variability of catchment characteristics across the study domain. Hydroclimatic attributes are computed for April/1987 – May/2020. The white star in panel (a) denotes the outlet of the Maipo en El Manzano River basin, for which the analysis approach is illustrated (see section 4.1).



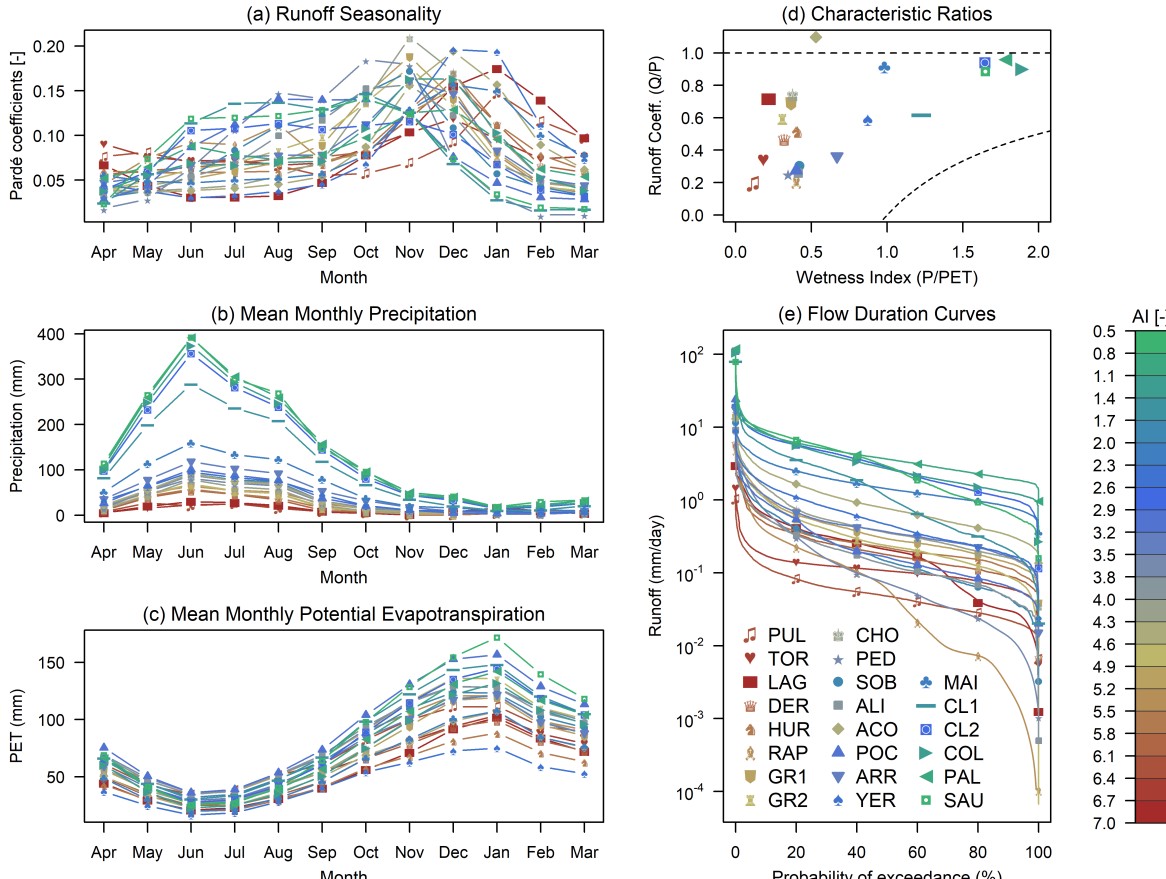


**Figure 2. Study basins' characteristics: (a) runoff seasonality, (b) mean monthly precipitation, (c) mean monthly potential evapotranspiration, (d) characteristic ratios, and (e) daily flow duration curves (FDC) for the period April/1987 – March/2020. In the legend (panel e), the basins are ordered from north (PUL) to south (SAU), and the colors indicate their aridity indices (AI; green to red – lower to higher index).**

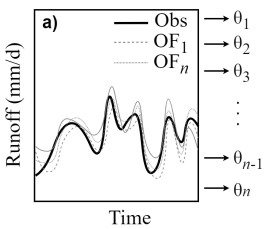
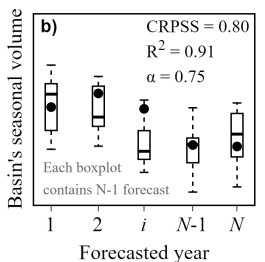
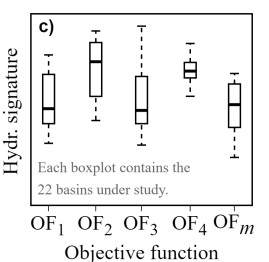
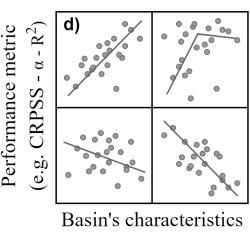

A hydrological model is calibrated and validated against observations (Obs) from 22 basins using *n* different objective functions (OF), resulting in *n* parameter sets ($\theta$)

Seasonal streamflow forecasts are produced with the ESP method and each parameter set for *N* years and five initialization times. The forecast for each season is produced by running the model with historical meteorology from the remaining *N-1* years. Several performance metrics are computed.

From **b)** the *m < n* best performing objective functions were selected. We examined hydrological consistency of simulated runoff for each of the *m* parameter sets

We explore possible controls (i.e., catchment attributes) that explain the performance of ESP hindcasts for different objective functions and initialization times.

**Figure 3.** Schematics of the approach used in this study. The steps are repeated for the 22 case study basins and three conceptual rainfall-runoff models.


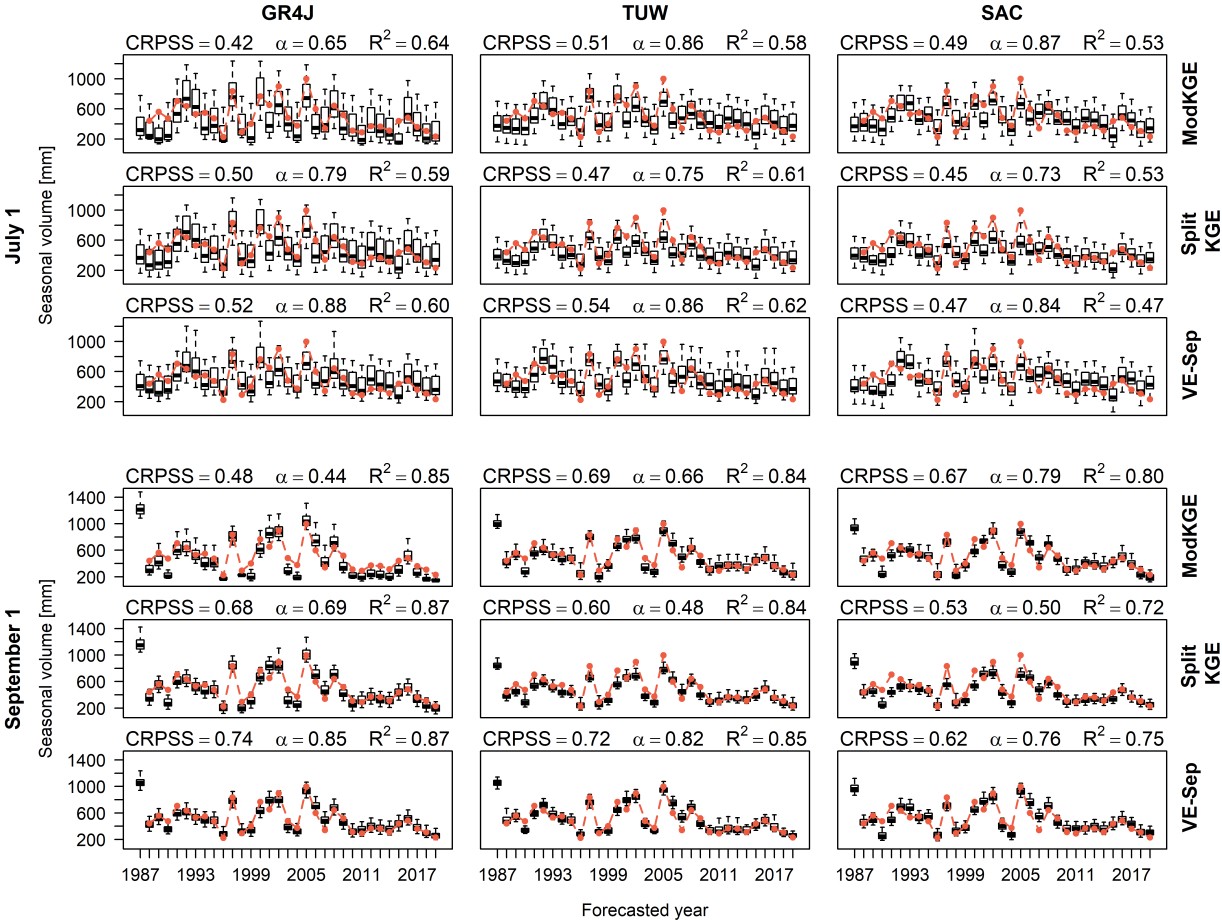

**Figure 4.** Time series with ESP seasonal hindcasts (i.e., September-March runoff) initialized on July 1, and September 1 for the Maipo at El Manzano basin. The boxes correspond to the interquartile range (IQR, i.e., 25th and 75th percentiles); the horizontal line in each box is the median, whiskers extend to the $\pm 1.5 \cdot IQR$ of the ensemble, and the red dots represent the observations. The results were produced with three model structures, using parameters obtained from calibrations conducted with ModKGE, Split KGE and VE-Sep (see details in Section 3.1). Each panel displays the CRPSS, the reliability index α, and the coefficient of determination R2 (computed using the ensemble forecast median).





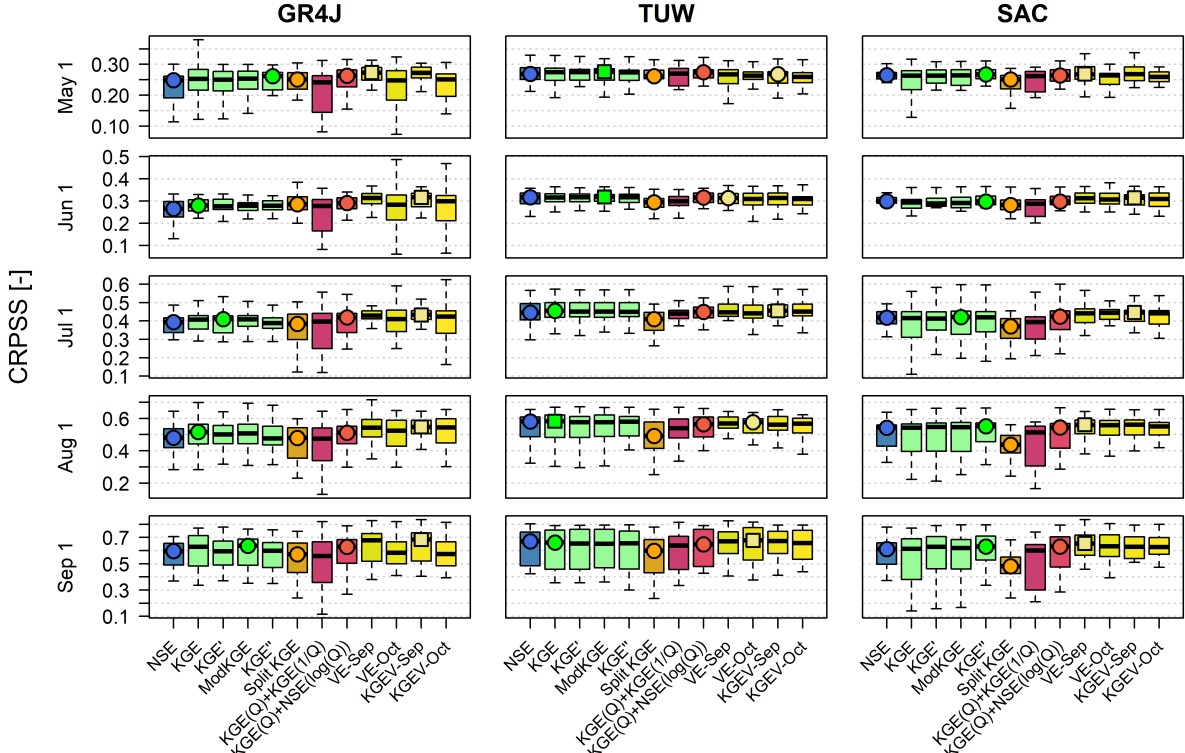

**Figure 5.** Comparison of CRPSS obtained with different calibration objective functions. Each panel contains results for a specific combination of initialization time (rows) and hydrological model (columns), and each boxplot comprises results from the 22 case study basins. The boxes correspond to the interquartile range (IQR, i.e., 25th and 75th percentiles), the horizontal line in each box is the median, and whiskers extend to the $\pm 1.5 \cdot IQR$ of the ensemble. The circle indicates the objective function providing the highest median within each family of calibration metric (identified with different colors), and the square indicates the objective function that delivers the best set of metric values using a specific combination of initialization time and hydrological model.





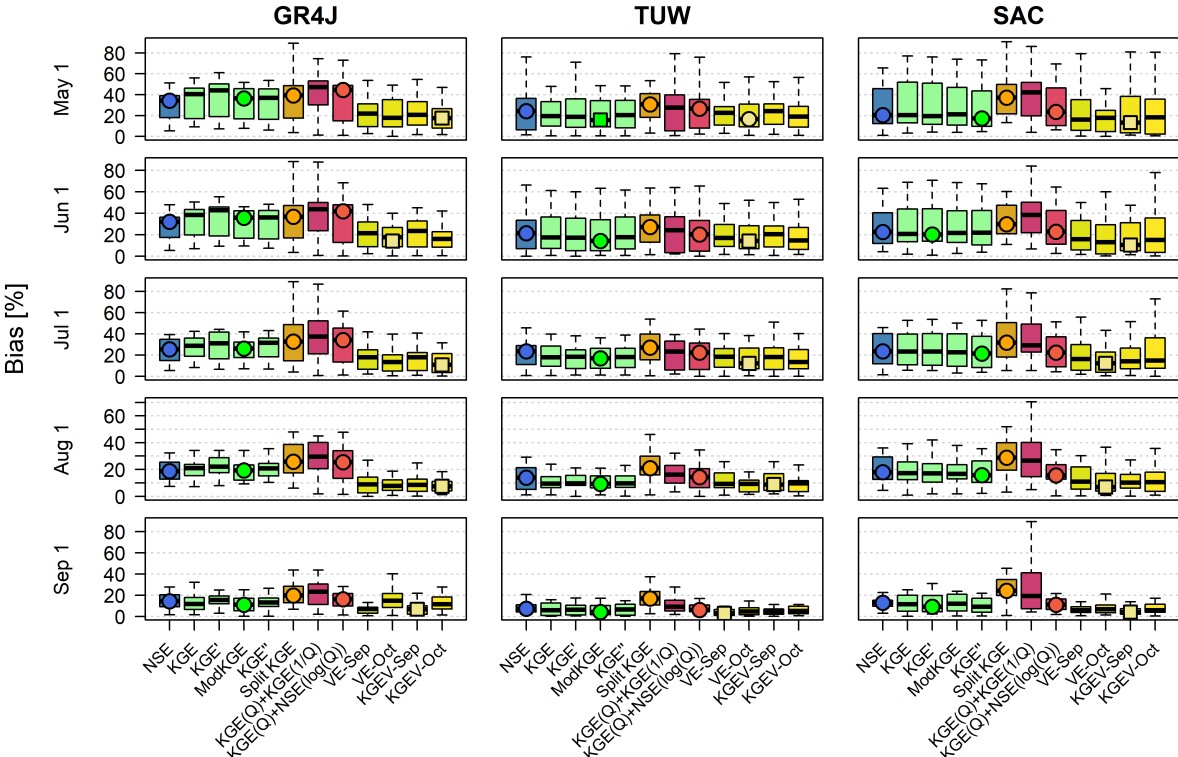


**Figure 6.** Same as in Figure 5, but for bias in absolute value.

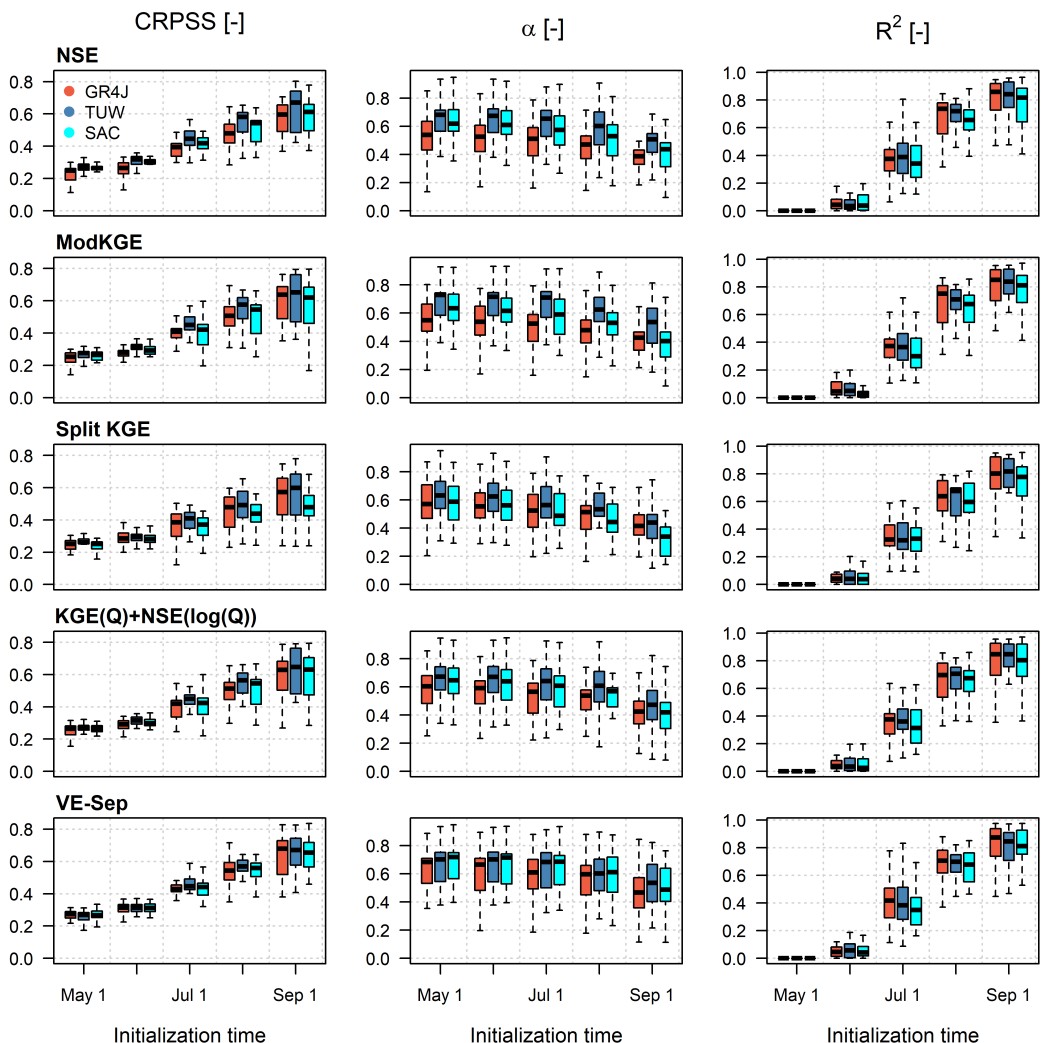

Figure 7. **Impact of initialization time on forecast verification metrics (columns) for the five representative calibration objective functions (rows). Each boxplot comprises results from the 22 case study basins and one hydrological model. The boxes correspond to the interquartile range (IQR, i.e., 25th and 75th percentiles), the horizontal line in each box is the median, and whiskers extend to the ±1.5·IQR of the ensemble.**



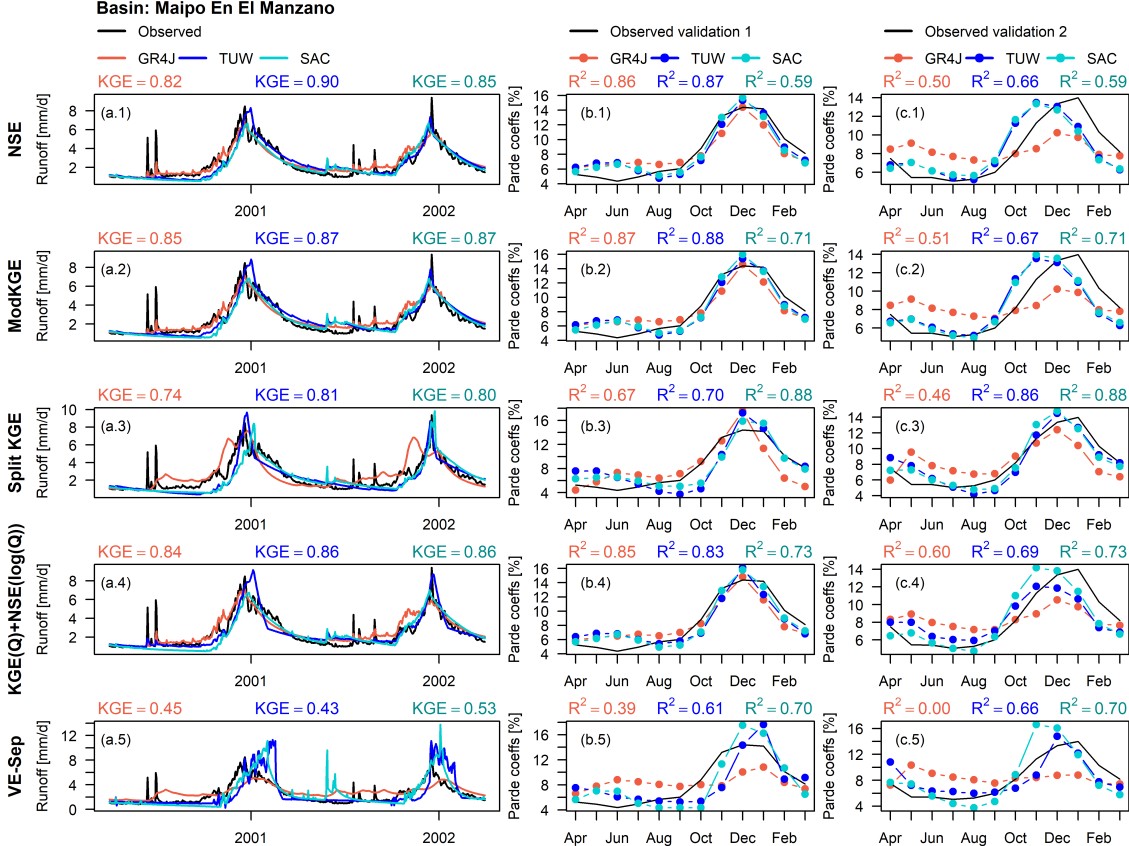

**Figure 8. (Left)** Daily hydrographs (April/2001 – March/2003) and seasonal variation curves for **(center)** the first (April/1987 – March/1994) and **(right)** second (April/2013 – March/2020) validation periods at the Maipo en el Manzano River basin, obtained with the three models and five representative objective functions: **(a)** NSE, **(b)** ModKGE, **(c)** Split KGE, **(d)** KGE(Q)+NSE(log(Q)) and **(e)** VE-Sep. The daily KGE (period April/2000-March/2002) obtained with each model is displayed in the left panels, while center and right panels include the coefficient of determination ($R^2$) between mean monthly simulated and observed runoff averages.



**Figure 9.** Percent biases in hydrologic signatures (rows) obtained with the five representative objective functions and the three hydrological models (shown in different colors) for the (left) calibration (April/1994 – March/2013), (center) verification 1 (April/1987 – March/1994) and (right) verification 2 (April/2013 – March/2020) periods. Each boxplot comprises results for our 22 case study basins. The boxes correspond to the interquartile range (IQR, i.e., 25th and 75th percentiles), the horizontal line in each box is the median, and whiskers extend to the $\pm 1.5 \cdot IQR$ of the ensemble. The values above each panel correspond to the median of the TUW model boxplots.



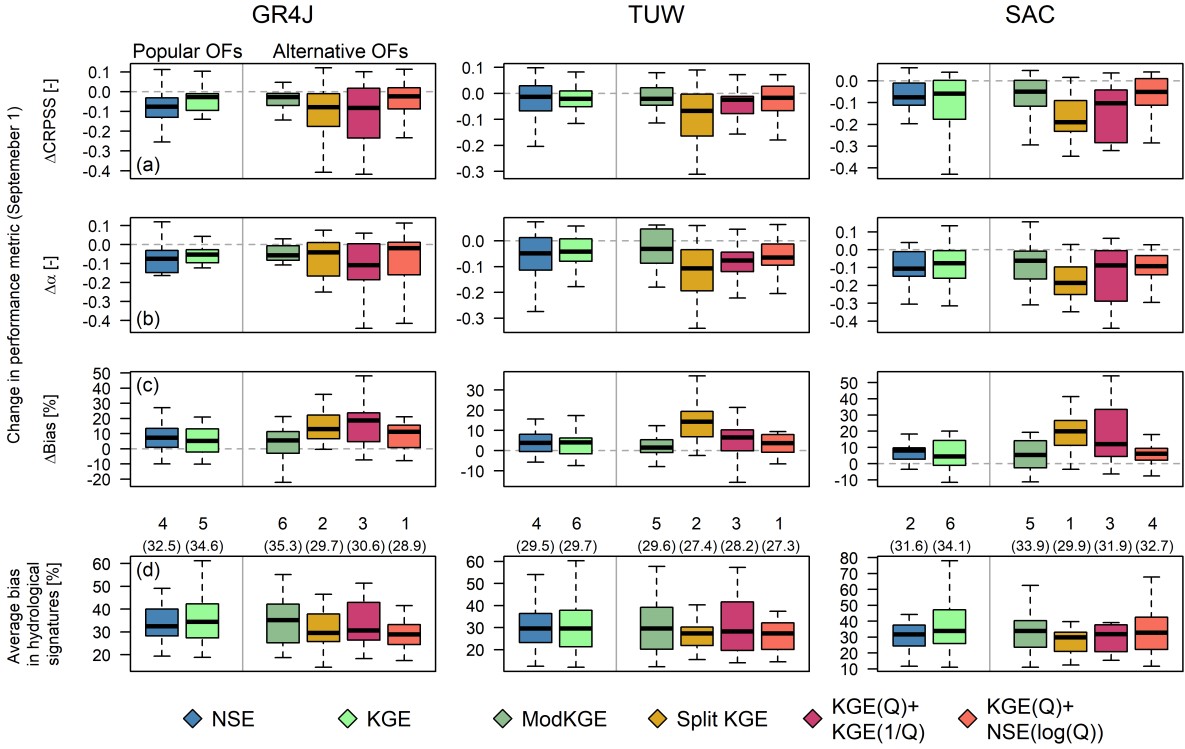


**Figure 10. Variations in September 1 CRPSS, α index and bias (panels (a) to (c)) due to the choice of popular and alternative objective functions (shown in different boxplots), relative to the best performing OF in terms of forecast quality (VE-Sep). The dashed line indicates no difference (i.e., no loss) in forecast performance. The bottom panels (d) display the average bias in hydrological signatures (computed over the calibration and two validation periods) with the associated ranking (being 1 the best in terms of hydrological consistency), and median average bias obtained from the sample of basins (in parentheses).**





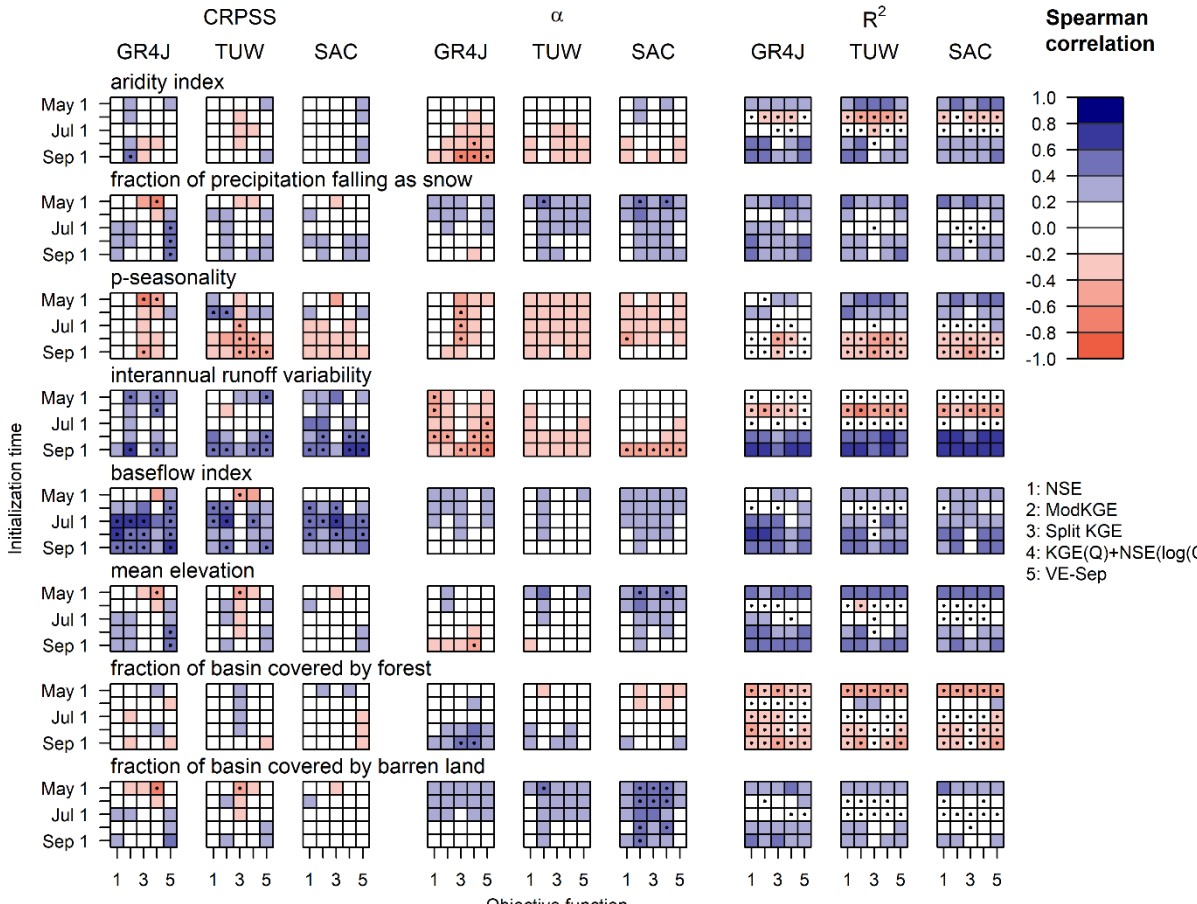

**Figure 11.** Spearman's rank correlation coefficients between catchment characteristics (shown in different rows) and the CRPSS (left), α reliability index (center), and the coefficient of determination $R^2$ (right) obtained for seasonal streamflow forecasts (period April/1987 – March/2020), produced with the five representative objective functions (x-axis in each color matrix), different initialization times (y-axis in each color matrix), and the three hydrological models. Black dots indicate statistically significant ($p < 0.05$) correlations.