# Peer review of "Towards robust seasonal streamflow forecasts in mountainous catchments: impact of calibration metric selection in hydrological modeling"

_Hydrology and Earth System Sciences, 2023_

## Author Comment (AC3)

Replies to reviews

**"Towards robust seasonal streamflow forecasts in mountainous catchments: impact of calibration metric selection in hydrological modeling"**

Diego Araya, Pablo A. Mendoza, Eduardo Muñoz-Castro and James McPhee

We thank the three reviewers for their time in commenting on our paper. We provide responses to each individual point below. For clarity, comments are given in italics, and our responses are given in plain blue text.

**Reviewer #3**

*Thank you for the opportunity to review this manuscript. The authors have clearly put substantial effort into this work to analyzing impact of calibration metrics, and providing potentially valuable insights in mountainous areas.*

We greatly appreciate the reviewer's positive feedback, as well as the time and consideration for providing constructive suggestions.

*While the overall presentation of the content is satisfactory, some of the arguments need stronger scientific support and detailed information. As other referees have already pointed out, the structure needs to be improved, while the content needs also to be filtered by relevance to reduce redundancy.*

In response to all the reviewers' comments, we have re-organized and re-designed most of the figures, reducing the amount of results included in the original version of the manuscript.

*In terms of specific comments:*

*Lines 20-25, please expand or explain abbreviations when they are first used to ensure clarity.*

We have added a small explanation for these metrics:
- Split KGE, which gives equal weight to each water year in the calibration time series.
- VE-Sep, which quantifies seasonal volume errors.

*The cited work "Troin et al., 2021" doesn't appear in the reference list.*

We have deleted this reference from the manuscript.

*Line 112, There is a typo error, should be "km2".*

Solved. Thanks!

*Line 123, could the authors expand on the underestimation they mention here? Is the underestimation from CR2MET or CAMELS dataset?*

In response to the reviewer's observation, we have added the following text in section 2:

"(…) Aconcagua at Chacabuquito (ACO) is the only basin with a mean annual runoff ratio larger than 1, which can be explained by (i) underestimation of catchment-averaged precipitation **from CR2MET or from the meteorological station records used to develop the gridded product**, (ii) **positive biases in streamflow records from the DGA's stations due to uncertainties in stage-discharge relationships**, or (iii) glacier and/or groundwater contributions.

*Line 123 and 126, similar as the previous comment: bias in runoff is mentioned in line 123, but it's unclear how this ties into the observed runoff mentioned in line 126. Further clarification would be useful.*

Please see our response to the previous comment.

*Could the authors clarify which dataset is used to conduct Figures 1 and 2?*

These graphs were produced using data retrieved from the CAMELS-CL database. We have added this information to the figure captions.

*On line 128, the manuscript mentions both basin-averaged precipitation from CR2MET and precipitation from CAMELS. Could the authors elaborate on how they are used? It would be helpful if they clarified the specific roles these data sources play in their analysis.*

CAMELS-CL contains daily time series of basin-averaged hydrometeorological variables retrieved from different sources, including CR2MET for the case of precipitation. To clarify this, we have added the following text in section 2:

"We use daily time series of observed streamflow, and basin-averaged precipitation, mean air temperature and potential evapotranspiration (PET) retrieved from the CAMELS-CL database (Alvarez-Garreton et al., 2018), which compiles information from different sources: (i) streamflow observations are acquired from stations maintained by the Chilean General Water Directorate (DGA), also available at the DGA's website (https://dga.mop.gob.cl/); (ii) basin-averaged precipitation and mean temperature data are derived from the gridded observational product CR2MET (DGA, 2017; Boisier et al., 2018) version 2.0, which provides information of these variables for continental Chile at a 0.05° x 0.05° horizontal resolution; and (iii) PET is calculated with the formula proposed by Hargreaves and Samani (1985) using basin averaged temperature from CR2MET."

*Line 240, I assume WY stands for Water Years? But this is not stated in the contents when they are first mentioned.*

Thank you for catching this. We now define the acronym WY in section 2.

*There are a large number of citations scattered throughout the paper, which makes it challenging to follow. Please consider revisiting these citations and remove any that may not be strictly necessary.*

We have removed citations where possible, following the reviewer's recommendation.

*Again, thank you for the opportunity to review this work. With clarifications and improvements, I believe this paper has potential to make a valuable contribution to the field.*

We are grateful to the referee for his/her thorough review and for providing valuable and constructive suggestions.

**References**

Alvarez-Garreton, C., Mendoza, P. A., Pablo Boisier, J., Addor, N., Galleguillos, M., Zambrano-Bigiarini, M., Lara, A., Puelma, C., Cortes, G., Garreaud, R., McPhee, J. and Ayala, A.: The CAMELS-CL dataset: Catchment attributes and meteorology for large sample studies-Chile dataset, Hydrol. Earth Syst. Sci., 22(11), 5817–5846, doi:10.5194/hess-22-5817-2018, 2018.

Boisier, J. P., Alvarez-Garretón, C., Cepeda, J., Osses, A., Vásquez, N. and Rondanelli, R.: CR2MET: A high-resolution precipitation and temperature dataset for hydroclimatic research in

Chile., 2018.

DGA: Actualización del balance hídrico nacional, SIT N°417, Ministerio de Obras Públicas, Dirección General de Aguas, División de Estudios y Planificación, Santiago, Chile., 2017.

Hargreaves, G. H. and Samani, Z. A.: Reference Crop Evapotranspiration from Temperature, Appl. Eng. Agric., 1(2), 96–99, doi:10.13031/2013.26773, 1985.

---

## Author Response (AR1)

Replies to reviews

**"Towards robust seasonal streamflow forecasts in mountainous catchments: impact of calibration metric selection in hydrological modeling"**

Diego Araya, Pablo A. Mendoza, Eduardo Muñoz-Castro and James McPhee

We thank the three reviewers for their time in commenting on our paper. We provide responses to each individual point below. For clarity, comments are given in italics, and our responses are given in plain blue text.

**Reviewer #1**

*The manuscript aims to evaluate the role of calibration metrics (objective function for calibration and performance evaluation metrics) on the seasonal streamflow forecasts in 22 mountainous river basins in Chile based on CAMELS-CL datasets. The quantum of work done by the authors needs appreciation, as well as the framing of the scientific questions. The manuscript has enough scientific content to be published in HESS after revision. The problems framed in the manuscript are tested using scientifically sound methodology.*

*However, I feel the manuscript is a complicated read due to the multiple parameters, metrics and lack of clarity, especially in the methods section. I believe the manuscript can benefit from reorganizing the content. The main result must be better highlighted, and others could be moved to the supplementary section to improve readability. The result sections do not highlight the overall conclusion or takeaway in each section. Therefore, I was pretty confused, even after multiple reads, about what the authors were trying to communicate.*

We greatly appreciate the reviewer's comments, and provide detailed responses below.

*A detailed flow chart can be used to convey the method. Parts of the methodology are distributed across different sections, including the introduction.*

We have designed a new and more detailed flow chart. Additionally, we have added a diagram inspired by Figure 3 in Crochemore et al. (2020) to explain the Ensemble Streamflow Prediction (ESP) method, which is now detailed in section 3.2 (L201-219).

[Figure]

**(a)** Calibration of hydrological models

**(b)** Ensemble Streamflow Prediction (ESP)

**(c)** Identification of robust objective functions (OFs)

**(d)** Relationship between hindcast performance and basin attributes

Figure 3. Flowchart describing the approach used in this study. See text for details.

*The concept of Ensemble Streamflow Prediction used in the study is defined in the introduction section. I would appreciate elaborating on it in the methodology section instead. The introduction section should better focus on existing gaps in the literature and highlight the need for the present study.*

We have moved the full description of the Ensemble Streamflow Prediction (ESP) method to section 3.2 ("Hindcast generation and verification"). We choose to maintain a brief description of the method in the introduction, only to highlight how our study contributes to the existing literature by exploring the impact of calibration metric selection on the quality of seasonal streamflow forecasts.

*There is a lack of consistency in the terms used, which makes it more confusing. For instance, though hindcasts are performed in the paper, at certain places, forecasts are used.*

In response to this observation and the comments from Reviewer #2:
- We have modified the term "forecasts" by "hindcasts" when referring to our methods and results, since this work presents an assessment of retrospective forecasts obtained from the application of different model calibration metrics.
- We use the term 'forecasts' when referring to past studies and operational applications.
- We use the term 'verification' when referring to the assessment of retrospective seasonal streamflow hindcasts.
- We use the term "evaluation" to the assessment of streamflow simulations outside the calibration period. The term "validation" is no longer used in this paper.

We have clarified this terminology at the beginning of section 3 in the revised manuscript (L136-139):

"In this paper, we use the term *forecast* when referring to past studies, applications at locations where observational data will not be available, and to reflect on the implications of our results for operational practice; we use the term *hindcast* when referring to retrospective forecasts produced in this study; the term *evaluation* for the assessment of streamflow model simulations outside the calibration period, and *verification* for the assessment of seasonal streamflow hindcasts."

*Similarly, I did not understand what the authors meant by the first and second validation periods in the Figure 8 caption. Did you mean the calibration period, where the model is calibrated using different parameters, and the hindcast period, where the ensemble streamflow prediction method is employed?*

As pointed above, we have replaced the term "validation" by "evaluation", which refers to the process of evaluating the quality of streamflow simulations outside the calibration period. Additionally, we have merged the originally proposed first and second evaluation periods into a single evaluation data set (which spans April/1987 – March/1994 and April/2013 – March/2020) to assess model simulations graphically and quantitatively. Such evaluation is illustrated for three objective functions in Figure 4 (former Figure 8). We clarify this procedure in Section 3.1.2 (L196-199):

"Model evaluation is conducted by computing performance metrics using data from two periods: (i) April/1987 - March/1994, which is hydroclimatically diverse, and (ii) April/2013 – March/2020, which is characterized by unprecedented and temporally persistent dry conditions (Garreaud et al., 2017, 2019). In both cases, the preceding 8-year period is used for model spin-up".

[Figure]

Figure 4. (Left) Daily hydrographs (April/2009 – March/2011) and (right) monthly variation curves for the evaluation dataset (April/1987 – March/1994 and April/2013 – March/2020) at the Maipo en el Manzano River basin, obtained with the three models and three objective functions: (1) NSE, (2) KGE(Q)+NSE(log(Q)) and (3) VE-Sep. The daily KGE obtained with each model is displayed in the left panels, while right panels include the coefficient of determination ($R^2$) between mean monthly simulated and observed runoff averages.

*I think the manuscript will also benefit from redesigning the figures. The multiple boxplot figures create a lot of complexity in analyzing. Reducing the amount of noise while focusing on a particular science question could considerably improve the manuscript's readability and merit.*

We appreciate this observation, and we agree that the figures included the original manuscript where unnecessarily busy. Hence, we have redesigned Figures 3 to 10, and we have moved Figure 6 (now Figure S5) in the original manuscript to Supplementary Material, following the recommendation of Reviewer #2.

*For instance, I would suggest focusing on the median result of all model combinations while showing the effect of initialization time and performance metrics for each calibration OF (Figure 7).*

In response to the comments from Reviewers #1 and #2, we have redesigned Figure 7, showing three hindcast performance metrics and five initialization times from only one model structure (TUW), with parameters calibrated with only one objective function (NSE). We display the median (solid line) and the 5th & 95th percentiles from the 22 case study basin basins as a light-blue shade for each metric. The results presented in this figure communicate the same findings obtained for the remaining representative objective functions and models: CRPSS and $R^2$ ($\alpha$-index) values increase (degrades) as hindcast initializations approach Sep. 1. We decided to keep all five initialization times to clearly show the progression of seasonal (i.e., September-March) hindcast quality during the austral winter. The extended version of the new Figure 6 (which contains all five representative objective functions) is now included in the Supporting Information document. We make all these points in section 4.2 of the revised manuscript (L296-303):

"Figure 7 illustrates how initialization time affects hindcast quality attributes when using NSE as calibration metric and the TUW model. As observed in the Upper Maipo River basin (**Error! Reference source not found.**), CRPSS and $R^2$ (the $\alpha$ index) improve (degrades) as hindcasts initializations approach September 1, with considerable increments in skill on July 1 compared to May 1 and June 1 hindcasts. The skill of May 1 hindcasts is rather low (with CRPSS 5th and 95th percentiles, obtained from the 22 catchments, equal to 0.26 and 0.28, respectively) and does not improve considerably on June 1. Additionally, inter-basin differences in CRPSS increase as hindcast initializations approach the beginning of the snowmelt season, ranging 0.57-0.69 on September 1. The same patterns, with small variations in ranges, are observed for the remaining representative objective functions and models (see Figures S7, S8 and S9 in Supporting Information)."

[Figure]

Figure 7. Impact of initialization time on (a) CRPSS, (b) the $\alpha$ reliability index, and (c) $R^2$ for seasonal streamflow hindcasts (period April/1987-March/2020) produced with the NSE as calibration objective function and the TUW model. The shades represent the 5th and 95th percentiles in each metric from the 22 case study basins, and the solid line represents the median value from the sample of catchments.

*Figure S2 is not cited, and supplementary figure S3 is wrongly numbered.*

We thank the reviewer for the detailed revision of our manuscript. Figure S2 is now cited in the revised manuscript (L272). Additionally, we have made sure that all the figures contained in the Supplementary material are correctly cited in the main manuscript.

*In respect of results, '(not shown)' is used multiple times in the manuscript. I would suggest it will be better to include it in the supplementary section if the results are an important part of the argument.*

We now include in the supplementary material most of the results that were referred to as 'not shown', in order to better support the arguments exposed here.

*I reiterate the scientific questions in the manuscript intend to improve the seasonal ensemble streamflow prediction by assessing its sensitivity to calibration metrics is an important question. However, improving the organization and presentability of results are required to understand the manuscript outcomes better.*

We agree with the Reviewer and thank him/her for the positive feedback and for his/her constructive review.

**Reviewer #2**

*This study evaluates the effect of calibration, initial condition and choice of model structure on forecasting seasonal volumes in mountainous areas. To that end, the parameters of three conceptual models were calibrated for 22 river basins in Chile using 12 objective functions. The calibrated models were then used to produce ESP forecasts considering five initial conditions. The authors evaluated the quality of the spring-summer forecasts using 33 years of data given the different combinations of models, parameters and initial conditions. An evaluation of the links between forecast quality, model performance in simulating different streamflow signatures, and catchment characteristics was conducted. The authors found that the choice of objective function has an impact on forecast quality and that a high performance in simulating hydrological signatures does not ensure good forecast quality. The authors also found that these results depend to some extent on the hydrological model.*

*As mentioned by the first reviewer, this study represents a significant amount of work. The subject is very relevant and has not been covered much in the literature. The differences in forecast quality and hydrological consistency for the different forecast combinations demonstrate that this study covers an important topic that needs to be considered for operational use.*

*Given the amount of model/calibration/HIC combinations and the choice of presentation by the authors, the article is not very easy to understand. I also have several questions and comments on some of the methodological choices that were made. Because the questions, comments and corrections I am making below might require a lot of work, I think they should be considered in light of the choices that will be made to reorganize the paper.*

We express our gratitude to the referee for his/her meticulous review of our manuscript and for the constructive suggestions. We provide our responses below.

*Major comments:*

*Twenty-two catchments with seasonal snowmelt contributions to total runoff were selected from the CAMELS-CL dataset for this study. It is stated (line 105) that "the selected basins are included in the CAMELS-CL dataset ... and meet the following criteria…". Were all the catchments meeting these criteria selected (resulting in a total of 22 catchments) or were there other mountainous catchments of CAMELS-CL meeting the same criteria? Although these catchments encompass a large variety of hydroclimatic conditions, a larger dataset would enable more general conclusions to be drawn from this study. If other catchments of CAMELS-CL were to meet the same criteria, I would suggest including them in this study. Since they come from the same open-source database, it would mean running the same calculations but for more catchments.*

Although we agree with the reviewer in that a larger sample of catchments would be desirable, there were no other Andean catchments meeting the six conditions. The most restrictive criteria are: (v) at least 75% of days with streamflow observations during the period April/1987 – March/2020, and (vi) at least 20 water years with seasonal (Sep-Mar) streamflow observations for hindcast verification purposes. We consider that these two requirements are essential for proper hydrologic model calibration and evaluation (since seasonal objective functions rely solely on Sep-Mar data availability) and a robust verification of seasonal streamflow hindcasts. We have added the following lines to section 2 to clarify this (L105-108):

"The most restrictive conditions are (v) and (vi), which hinder the possibility to include additional mountainous catchments from CAMELS-CL; nevertheless, we consider that both requirements are essential for proper hydrologic model calibration and evaluation (since seasonal objective functions rely solely on Sep-Mar data availability) and a robust verification of seasonal streamflow hindcasts."

*If the choice of selecting 22 catchments was driven by computation time restrictions, please consider mentioning it in the manuscript.*

The choice of catchments was not driven by computation time restrictions. Please see our previous response.

*As pointed out by the first reviewer, the manuscript can benefit from reorganizing the content. One way to reorganize the content would be to keep only one figure presenting the results for all models/objective functions/HICs (e.g. fig. 5, extended in height to enhance visualisation) and change the other figures so that they better highlight the conclusions of the paper.*

We agree with the reviewers that the original set of figures has a high degree of complexity. Hence, we have redesigned Figures 3 to 10 to make them easier to understand and to better highlight the key results and conclusions of this work. Following the reviewer's recommendation, we now have only one figure (i.e., Figure 6, which contains CRPSS values) presenting the results for all models, all (12) calibration objective functions and hindcast initializations. Additionally, we have improved the methodology flow chart (Figure 3), and offer more detailed explanations in the text about the approach and results.

*A few related suggestions:*

*Fig. 4 and section 4.1: I would have put this section at the end of the results section to illustrate the results. This figure could be reduced in terms of results by only keeping one model and two OFs (a popular OF and KGE(Q)+NSE(log(Q)))*

We appreciate the reviewer's suggestion. However, we believe that it is important to illustrate first the individual basin calibration (Figure 3a) and the hindcast generation/verification (Figure 3b) steps in one case study basin. The extension of steps 3a and 3b (Figure 3) to the 22 case study catchments is the basis for subsequent analyses aiming to explore connections between seasonal hindcast performance, the hydrological consistency of streamflow simulations obtained with the various calibration metrics, and catchment characteristics. We now clarify this at the beginning of section 3 (L140-145).

"Figure 3 outlines our methodology, which includes four steps: (a) parameter calibration of three hydrological models (GR4J, TUW and SAC-SMA) configured in 22 snow-influenced basins using a suite of 12 objective functions; (b) seasonal (September-March) streamflow hindcast generation with the ESP method for 33 WYs (April/1987 - March/2020) and five initialization times, and verification of forecast quality attributes; (c) assessment of hydrological consistency through five streamflow signatures for the subset of best-performing objective functions in terms of hindcast attributes, and (d) analysis of possible relationships between catchment characteristics and ESP hindcast attributes."

Additionally, we have simplified the original Figure 8 (as also suggested by Reviewer #1) and placed it before the original Figure 4 (now Figure 5). These results are included in Section 4.1, which has been renamed to "Example: hydrologic model calibration and ESP at the Upper Maipo River basin". Nevertheless, we fully agree that the information content of the original Figure 4 was excessive, and have hence simplified it from a 6 x 3 panel figure, to a 3 x 2 panel figure. We now show results from one model structure (TUW model) and three calibration metrics: NSE (a popular OF as the reviewer suggests), KGE(Q)+NSE(log(Q)), and VE-Sep, due to their relevance in the subsequent results and findings.

[Figure]

Figure 5. Time series with ESP seasonal hindcasts (i.e., September-March runoff) initialized on July 1 (left panels), and September 1 (right panels) for the Maipo at El Manzano basin. The boxes correspond to the interquartile range (IQR, i.e., 25th and 75th percentiles); the horizontal line in each box is the median, whiskers extend to the ±1.5·IQR of the ensemble, and the red dots represent the observations. The results were produced with the TUW model, using parameters obtained from calibrations conducted with NSE, KGE(Q)+NSE(log(Q)) and VE-Sep (see details in Section 3.1). Each panel displays the CRPSS, the reliability index α, and the coefficient of determination R2 (computed using the ensemble forecast median).

*Fig. 5: to extend in height to enhance visualisation.*

We have modified this figure (now Figure 6) following the reviewer's suggestion. We have also changed the colours to enhance visualization.

[Figure]

Figure 6. Comparison of CRPSS obtained with different calibration objective functions. Each panel contains results for a specific combination of initialization time (rows) and hydrological model (columns), and each boxplot comprises results from the 22 case study basins. The boxes correspond to the interquartile range (IQR, i.e., 25th and 75th percentiles), the horizontal line in each box is the median, and whiskers extend to the ±1.5·IQR of the ensemble. The circle indicates the objective function providing the highest median within each family of calibration metric (identified with different colors), and the square indicates the objective function that delivers the best set of metric values using a specific combination of initialization time and hydrological model.

*Fig. 6: to move to the supplementary materials.*

We have moved this figure to the Supporting Information section (Figure S5), following the reviewer's recommendation.

*Fig. 7: show only three OFs (pick the most relevant to highlight your conclusions), May 1 and Sep 1 for initialization times and two forecast criteria.*

In response to the comments from Reviewers #1 and #2, we have redesigned Figure 7, showing three hindcast performance metrics and five initialization times from only one model structure (TUW), with parameters calibrated with only one objective function (NSE). We display the median (solid line) and the 5th & 95th percentiles from the 22 case study basin basins as a light-blue shade for each metric. We believe that the results from one model and one calibration metric are enough to communicate the same findings obtained for the remaining representative OFs and models: CRPSS and $R^2$ ($\alpha$-index) values increase (degrades) as hindcast initializations approach Sep. 1. We decided to keep all five initialization times to clearly show the progression of seasonal (i.e., September-March) hindcast quality during the austral winter. The extended version of the new Figure 6 (which contains

all five representative objective functions) will be included in the Supporting Information. We make all these points in section 4.2 of the revised manuscript (L296-303):

"Figure 7 illustrates how initialization time affects hindcast quality attributes when using NSE as calibration metric and the TUW model. As observed in the Upper Maipo River basin (**Error! Reference source not found.**), CRPSS and $R^2$ (the α index) improve (degrades) as hindcasts initializations approach September 1, with considerable increments in skill on July 1 compared to May 1 and June 1 hindcasts. The skill of May 1 hindcasts is rather low (with CRPSS 5th and 95th percentiles, obtained from the 22 catchments, equal to 0.26 and 0.28, respectively) and does not improve considerably on June 1. Additionally, inter-basin differences in CRPSS increase as hindcast initializations approach the beginning of the snowmelt season, ranging 0.57-0.69 on September 1. The same patterns, with small variations in ranges, are observed for the remaining representative objective functions and models (see Figures S7, S8 and S9 in Supporting Information)."

[Figure]

Figure 7. Impact of initialization time on (a) CRPSS, (b) the $\alpha$ reliability index, and (c) $R^2$ for seasonal streamflow hindcasts (period April/1987-March/2020) produced with the NSE as calibration objective function and the TUW model. The shades represent the 5th and 95th percentiles in each metric from the 22 case study basins, and the solid line represents the median value from the sample of catchments.

*Fig. 8: keep only two OFs and present the daily hydrographs for two years in one of the validation periods.*

In response to the recommendations provided by reviewers #1 and #2, we have merged the originally proposed first and second evaluation periods into a single evaluation data set (which spans April/1987 – March/1994 and April/2013 – March/2020). Hence, we show daily hydrographs for two water years from the evaluation data set, and the runoff seasonality curves are displayed jointly for these two periods. We decide to show results for three representative objective functions (the same as in Figure 5), and place this figure before.

[Figure]

Figure 4. (Left) Daily hydrographs (April/2009 – March/2011) and (right) monthly variation curves for evaluation data set (April/1987 – March/1994 and April/2013 – March/2020) at the Maipo en el Manzano River basin, obtained with the three models and three objective functions: (1) NSE, (2) KGE(Q)+NSE(log(Q)) and (3) VE-Sep. The daily KGE obtained with each model is displayed in the left panels, while right panels include the coefficient of determination ($R^2$) between mean monthly simulated and observed runoff averages.

*Fig. 9: keep only two or three OFs and extend the figure in height.*

We have extended the figure (now numbered as Figure 8) in height and have simplified it by removing one of the signatures (F9M, since it contained very similar information of high flows than FHV), and showing the results from only one model (TUW). Nevertheless, we decided to keep the five representative objective functions for consistency with Figure 6, which highlights them as the best performing OFs per family. Additionally, we now show percent biases in hydrological signatures for the calibration period and for the combined evaluation data set, which combines the periods April/1987 – March/1994 and April/2013 – March/2020.

[Figure]

Figure 8. Percent biases (y-axis) in hydrologic signatures (x-axis) obtained with the five representative objective functions and the TUW model for the (a) calibration (April/1994 – March/2013) and (b) evaluation dataset (April/1987 – March/1994 and April/2013 – March/2020). Each boxplot comprises results for our 22 case study basins. The boxes correspond to the interquartile range (IQR, i.e., 25th and 75th percentiles), the horizontal line in each box is the median, and whiskers extend to the $\pm 1.5 \cdot IQR$ of the ensemble.

*Fig. 10: remove the alpha and bias lines. Show only KGE or NSE for popular OFs.*

We have modified this figure (now numbered as Figure 9) following the reviewer's suggestion. We have also removed the results from the KGE(Q)+KGE(1/Q) objective function because it did not add any relevant information relative to what is illustrated with the remaining objective functions.

[Figure]

Figure 9. Variations in September 1 CRPSS (top panels) due to the choice of popular and alternative objective functions (shown in different boxplots), relative to the best performing OF in terms of forecast quality (VE-Sep). The dashed line indicates no difference (i.e., no loss) in forecast performance. The bottom panels display the average bias in hydrological signatures (computed over the calibration period and evaluation data set) with the associated ranking (being 1 the best in terms of hydrological consistency), and median average bias obtained from the sample of basins (in parentheses).

*I did not understand why the ESP forecasts were evaluated on both calibration and evaluation periods without separation in the result analyses. Even if there are 32 meteorological inputs for each forecasted year, and that they are different from the meteorological input of the forecasted year, the streamflow data used to calculate seasonal performance has already been "seen" by the model during calibration. Since one of the goals of this study is to evaluate the impacts of calibration on seasonal forecasts, I think the authors should consider the evaluation of forecast quality only on the evaluation periods (or better explain why it was done that way). 33-1 years of meteorological data can still be used for the ensemble forecasts of each forecasted year. To improve the analysis of the hydrological consistency of model simulations, the authors could perform a split-sample test.*

We agree with the reviewer in that, when the hindcasted year overlaps with the calibration period (as it happens with our experimental setup), the hydrological model gains information from meteorological inputs, even if the climate time series observed during that year are excluded from the generation of ESP hindcasts. In spite of this, we decided to take advantage of the entire 33-year period for hindcast verification, since small sample sizes (i.e., number of WYs) have been widely recognized as a serious limitation within the seasonal forecasting literature (e.g., Shi et al., 2015; Trambauer et al., 2015; Mendoza et al., 2017; Lucatero et al., 2018; Wood et al., 2018). This strategy enables a more robust assessment of seasonal hindcast quality, as opposed to using only the 14 WYs left for model evaluation. To demonstrate this point, we characterized the impact of sample size on the spread of CRPSS results by performing a bootstrap analysis with 1000 realizations for the Maipo River basin, using hindcasts produced with the TUW model and KGE(Q)+NSE(log(Q)) as the calibration metric (Figure 11). The analysis was conducted for the following verification samples: (a) full period (i.e., 33 WYs) using the same parameter set, obtained by calibrating the model with data from the period April/1994 – March/2013; (b) full period, using parameter sets re-calibrated with all data except the

hindcasted year (i.e., 33 parameter sets to produce 33 seasonal hindcasts); (c) 19 WYs (calibration periods), using a single parameter set obtained with data from the same period; (d) 14 WYs (i.e., evaluation data set April/1987 – March/1994 and April/2013 – March/2020), using the same parameter set as in case (c); and (e) 14 WYs (April/2006 – March/2020), using the same parameter set as in case (c).

The results in Figure 11 show a considerable spread in CRPSS arising from sampling uncertainty when using 14-year verification periods (orange and cyan boxes). Additionally, the median CRPSS results are lower than those obtained with 19 and 33 WYs in July 1, August 1 and September 1. An interesting result is the similarity of CRPSS values obtained with scenarios (a) and (b), suggesting that the hindcasting generation and verification approach adopted here (i.e., using a single parameter set obtained by calibrating will all the years with available observations) is a good proxy to characterize the hindcast quality that would be obtained with an operational setup that considers parameter re-calibration for each forecasted season.

We thank the reviewer for this thoughtful observation, and we have incorporated this analysis in the new section 5.5 ("Verification sample size"), contained in the discussion section (L421-441).

[Figure]

Figure 11: Comparison of CRPSS values for seasonal (i.e., September-March) streamflow hindcasts produced at the Maipo River basin with the TUW model and KGE(Q)+NSE(log(Q)) as calibration metric. Each box comprises results from 1000 bootstraps with replacement applied to different verification sample sizes (i.e., number of hindcast-observation pairs): (a) full period (i.e., 33 WYs) using the same parameter set, obtained by calibrating the model with data from the period April/1994 – March/2013 (blue); (b) full period, using parameter sets re-calibrated with all data except the hindcasted year (i.e., 33 parameter sets to produce 33 seasonal hindcasts, gray); (c) 19 WYs (calibration periods), using a single parameter set obtained with data from the same period (red); (d) 14 WYs (i.e., evaluation data set April/1987 – March/1994 and April/2013 – March/2020), using the same parameter set as in case (c) (orange); and (e) 14 WYs (April/2006 – March/2020), using the same parameter set as in case (c) (cyan). The boxes correspond to the interquartile range (IQR, i.e., 25th and 75th percentiles); the horizontal line in each box is the median, and the whiskers extend to the $\pm 1.5 \cdot IQR$ of the ensemble.

*Lines 21 and 435, the term "hydrologically consistent parameter set" is used. No analyses of parameter sets were made in this study, only the ability of the models to reproduce streamflow signatures was evaluated. A high performance for specific streamflow signatures may imply more*

*consistency in simulating streamflow than using a "popular" metric. However, I would argue that it does not necessarily imply that the parameter sets of the models are more hydrologically consistent, as a model can be wrongly parameterized and the hypotheses behind not fit for the studied catchments. Even when a specific parameter set leads to better signature performance, equifinality of parameters can remain high, especially if the parameter sets yielding good performance vary between periods. As the manuscript already includes many results, I suggest considering a small analysis of the parameter sets of one of the models (e.g. TUWmodel that seems to be giving the highest forecast quality). This analysis would only be conducted for three objective functions (the one associated with the lowest hydrological consistency, the one associated with the highest hydrological consistency and KGE(Q)+NSE(log(Q)) which is the best compromise between forecast quality and hydrological consistency). In TUW model, not all the parameters would need to be assessed but, for instance, only the ones related to baseflow (in the TUWmodel package, it would be "param k2") or/and snow, to relate the results to catchment attributes that have a strong correlation with forecast quality.*

We agree with the reviewer's appreciation and, therefore, we have removed any references to "hydrologically consistent parameter sets" in the revised manuscript, replacing by "hydrologically consistent simulations", as the reviewer suggests below. Despite we recognize that parameter equifinality can be substantial, characterizing such effects is out of the scope of this study. Recently, Muñoz-Castro et al. (2023) examined the effects of calibration metric selection and parameter equifinality on the level of (dis)agreement in parameter values across 95 catchments in Chile, finding that (i) the choice of objective function has smaller effects on parameter values in catchments with low aridity index and high mean annual runoff ratio, in contrast to dryer climates, and (ii) catchments with better parameter agreement also provide better performance across model structures and simulation periods. Future work could explore whether such performance in streamflow simulations translates well into seasonal forecast quality attributes. We will make these points in section 5.6 ("Limitations and future work") of the revised manuscript (L443-450).

*The variations of parameters between periods could then be evaluated (if you were to follow the previous comment about periods of calibration and evaluation).*

We appreciate the reviewer's sentiment. Nevertheless, the assessment of temporal stability in hydrological model parameters is itself a topic worth of detailed investigation (e.g., Merz et al., 2011; Coron et al., 2012; Gharari et al., 2013; Fowler et al., 2018; Duethmann et al., 2020) that we prefer to leave for future work. We make this point in section 5.6 ("Limitations and future work") of the revised manuscript (L450-454):

"Additionally, calibration strategies (e.g., Gharari et al., 2013; Fowler et al., 2018) and model selection frameworks (e.g., Saavedra et al., 2022) advocating for consistent performance across different hydroclimatic conditions could be explored for seasonal forecasting applications."

*Of course, this would need to be considered after a reorganisation of the manuscript (to better highlight the results already presented). It should not come in the same format as the other results. The content of the manuscript (in terms of results) needs to be reduced first. That being said, the publications needs some extra work. The main points that need attention are argumentation for hydrological model aggregation, the structure of text and figures, additional reflection on the meaning of study results, and the archiving of code and data.*

We have re-organized and re-designed most of the figures, reducing the amount of results included in the original submission, and we have revised the text accordingly. The data and codes used to produce the results presented here are correctly archived in a Zenodo repository (see our response below).

*Minor comments:*

*The link to the Zenodo repository does not seem to be working anymore (last checked: 30/06).*

The Zenodo repository is working again (last checked: August 15, 2023). We attach the message from Zenodo Support.

Dear Diego

In Zenodo we have automated mechanisms to block spam, however this system can sometimes make mistakes that lead to an issue with user blocking. We apologize for the inconvenience it has caused. Your account has been fully restored, and we have taken the appropriate measures to ensure it won't happen again.

Best regards,
Lars

Zenodo Support
https://zenodo.org

*Lines 27 and 86: the term "for the right reasons" is a bit strong, as no additional data to streamflow was used for model evaluation. I suggest using "more hydrologically consistent simulations" as you did in the remaining of the paper.*

We have deleted any reference to "the right reasons", and now refer to "hydrologically consistent simulations", following the reviewer's recommendation.

*Line 108: how were the seasonal snowmelt contributions calculated?*

We have deleted any reference to snowmelt contributions, since we did not estimate these. What we actually meant for criteria (iv) is that the selected basis have the requirement of snowmelt influence on runoff seasonality (i.e., they must have snow-driven, nivo-pluvial or pluvio-nival regimes, as described by Baez-Villanueva et al., 2021). We have modified the text in Section 2 to clarify this (L102-103).

*Line 155: the CemaNeige model also partitions total precipitation into liquid and solid precipitation. Liquid precipitation and snowmelt are fed to the soil moisture store.*

In response to the reviewer's observation, we have re-worded the text in section 3.1.1 as follows (L59-161):

"(…) The CemaNeige module first partitions total precipitation into liquid and solid, and then simulates snow accumulation and melt over five or more (user-defined; here we use 10) elevation bands, using a two-parameter degree-day based scheme (Valéry et al., 2014) that adds snowmelt and liquid precipitation to the soil moisture accounting reservoir. (…)".

*Line 158: the GR4J model also includes a non-conservative function for water exchanges between topographical catchments.*

In response to the reviewer's observation, we have added the following text in section 3.1.1 (L163-164):

"A groundwater exchange term acts on both flow components to represent water exchanges between topographical catchments."

*Line 160: what do you mean by response area?*

We meant response routine. We have modified the text to clarify (L168).

*Sect 3.1.1: were different elevation bands considered in TUWmodel and SNOW17?*

No. These models were implemented in a lumped fashion. We clarify this at the end of section 3.1.1 (L182-183):

"While the CemaNeige is configured with 10 elevation bands, the snow routines of TUW and SAC-SMA (i.e., SNOW-17) are implemented in a lumped fashion."

*Sect 3.1.1: the three models used in this study were implemented within the R environment. These models and their exact implementation were described in (Astagneau et al., 2021; https://doi.org/10.5194/hess-25-3937-2021). In addition, the structures of the models are compared using a unified representation of the different storages and fluxes (Fig. 1). If, and only if, you used this paper to choose, implement or understand these models, you should cite it. Otherwise, please ignore this comment.*

We did not select the model structures based on the paper mentioned by this reviewer. Instead, these model structures were selected because they are widely used by the hydrology community (Addor and Melsen, 2019), with a myriad applications to streamflow forecasting. For example, SAC-SMA has been applied for testing alternative approaches (e.g., Mendoza et al., 2017), and is used to produce operational streamflow forecasts in the US (Micheletty et al., 2021). GR4J has been applied to assess streamflow forecasting frameworks in large samples of catchments (e.g., Harrigan et al., 2018; Woldemeskel et al., 2018). HBV-like conceptual models have been used to assess short (e.g., Pauwels and De Lannoy, 2009; Verkade et al., 2013) to long (e.g., Peñuela et al., 2020) range streamflow forecasts, especially in European countries. This explanation is included in section 3.1.1 (L151-157).

*Fig. 3 and sect. 3.1.2: I am not sure there is any validation of the models made in this study. For me, validation means that you are choosing one model over the other (or rejecting one) and evaluation means that you are evaluating the models outside the calibration period. Please consider replacing "validation" by "evaluation".*

In response to the reviewer's observation, we have replaced the term "validation" by "evaluation" in the text and the figures. Here, evaluation refers to the assessment of streamflow model simulations outside the calibration period (see L138).

*Sect 3.2: it could be useful to add a more detailed presentation of the ESP method (for instance by adding a reference to Fig. 3 of Crochemore et al., 2020; https://doi.org/10.1029/2019WR025700) and extend Fig. 3 that really helps to understand your framework.*

We have designed a new and more detailed flow chart to explain the methodology. Additionally, we have added a diagram inspired by Figure 3 in Crochemore et al. (2020) to explain the Ensemble Streamflow Prediction (ESP) method.

[Figure]

Figure 3. Flowchart describing the approach used in this study. See text for details.

**Reviewer #3**

*Thank you for the opportunity to review this manuscript. The authors have clearly put substantial effort into this work to analyzing impact of calibration metrics, and providing potentially valuable insights in mountainous areas.*

We greatly appreciate the reviewer's positive feedback, as well as the time and consideration for providing constructive suggestions.

*While the overall presentation of the content is satisfactory, some of the arguments need stronger scientific support and detailed information. As other referees have already pointed out, the structure needs to be improved, while the content needs also to be filtered by relevance to reduce redundancy.*

In response to all the reviewers' comments, we have re-organized the material and re-designed most of the figures, reducing the amount of results included in the original submission, and we have revised the text accordingly.

*In terms of specific comments:*

*Lines 20-25, please expand or explain abbreviations when they are first used to ensure clarity.*

We have added a small explanation for these metrics:
- Split KGE, which gives equal weight to each water year in the calibration time series (L20).
- VE-Sep, which quantifies seasonal volume errors series (L22).

*The cited work "Troin et al., 2021" doesn't appear in the reference list.*

We have deleted this reference from the manuscript.

*Line 112, There is a typo error, should be "km2".*

Solved. Thanks!

*Line 123, could the authors expand on the underestimation they mention here? Is the underestimation from CR2MET or CAMELS dataset?*

In response to the reviewer's observation, we have added the following text in section 2 (L130-133):

"(…) Aconcagua at Chacabuquito (ACO) is the only basin with a mean annual runoff ratio larger than 1, which can be explained by (i) underestimation of catchment-averaged precipitation from CR2MET v2.0 or from the meteorological station records used to develop the gridded product, (ii) positive biases in streamflow records from the DGA's stations due to uncertainties in stage-discharge relationships, or (iii) glacier and/or groundwater contributions.

*Line 123 and 126, similar as the previous comment: bias in runoff is mentioned in line 123, but it's unclear how this ties into the observed runoff mentioned in line 126. Further clarification would be useful.*

Please see our response to the previous comment.

*Could the authors clarify which dataset is used to conduct Figures 1 and 2?*

These graphs were produced using data retrieved from the CAMELS-CL database. We have added this information to the figure captions.

*On line 128, the manuscript mentions both basin-averaged precipitation from CR2MET and precipitation from CAMELS. Could the authors elaborate on how they are used? It would be helpful if they clarified the specific roles these data sources play in their analysis.*

CAMELS-CL contains daily time series of basin-averaged hydrometeorological variables retrieved from different sources, including CR2MET for the case of precipitation. To clarify this, we have added the following text in section 2 (L109-115):

"We use daily time series of observed streamflow, and basin-averaged precipitation, mean air temperature and potential evapotranspiration (PET) retrieved from the CAMELS-CL database (Alvarez-Garreton et al., 2018), which compiles information from different sources: (i) streamflow observations acquired from stations maintained by the Chilean General Water Directorate (DGA), also available at the DGA's website (https://dga.mop.gob.cl/); (ii) basin-averaged precipitation and mean temperature data derived from the gridded observational product CR2MET (DGA, 2017; Boisier et al., 2018) version 2.0, which provides information of these variables for continental Chile at a 0.05° x 0.05° horizontal resolution; and (iii) PET calculated with the formula proposed by Hargreaves and Samani (1985) using basin averaged temperature from CR2MET."

*Line 240, I assume WY stands for Water Years? But this is not stated in the contents when they are first mentioned.*

Thank you for catching this. We now define the acronym WY in section 2 (L104).

*There are a large number of citations scattered throughout the paper, which makes it challenging to follow. Please consider revisiting these citations and remove any that may not be strictly necessary.*

We have removed citations where possible, following the reviewer's recommendation.

*Again, thank you for the opportunity to review this work. With clarifications and improvements, I believe this paper has potential to make a valuable contribution to the field.*

We are grateful to the referee for his/her thorough review and for providing valuable and constructive suggestions.

**References**

[revised manuscript text omitted]

---

## Author Response (AR2)

Replies to review

**"Towards robust seasonal streamflow forecasts in mountainous catchments: impact of calibration metric selection in hydrological modeling"**

Diego Araya, Pablo A. Mendoza, Eduardo Muñoz-Castro and James McPhee

We thank the reviewer for their time in commenting on our paper. We provide responses to each individual point below. For clarity, comments are given in italics, and our responses are given in plain blue text.

**Reviewer #2**

*The authors have addressed most of my comments. The manuscript is easier to read and the main conclusions are better highlighted by the new figures. I have one last major comment that I would like the authors to respond to, and a couple of minor comments.*

*Major comment:*

*I would like to thank the authors for responding to my comments regarding the distinction between calibration and evaluation periods for the analysis of hindcast performance. I realise that the main idea of my comments may not have been entirely clear, and I apologise for that. Although I agree that using the longest available period is important to obtain reliable hindcasts, I still think that there needs to be a distinction between evaluation and calibration periods for the hindcast performance analysis, as is done for the hydrological consistency analysis. As I understand it, one of the aims of this study is to evaluate the differences in hindcast performance and hydrological consistency for different objective functions and initialisation times. So, the main point is not just to get good hindcast performance - which in fact would require the use of some sort of threshold to determine what is acceptable performance - but to compare hindcast performance between different modelling setups. So I think the question is whether you would still get the same rankings in terms of the best objective function for hindcast performance if you had distinguished the evaluation from the calibration periods. For example, I would like to know if the VE-sep objective function is still the best objective function in terms of CRPSS when looking at the evaluation period. One solution to determine whether this changes the results in terms of objective function ranking could be to produce a very simple figure for the TUW model, 1 September CRPSS and the four objective functions of Figure 9, but with the distinction between evaluation and calibration periods. This could be added to Section 5.5 and would either replace or complement Figure 11.*

We appreciate the reviewer's comment, since it enriches our discussion on sample size issues in hindcast verification, especially when developing and testing seasonal forecasting methods. To address this observation, we examined the sensitivity of the CRPSS for September 1 hindcasts, to the stratification of the full verification sample (i.e., 33 WYs) between hydrologic model calibration (April/1994 – March/2013; i.e., 19 WYs) and evaluation (April/1987 – March/1994 and April/2013 – March/2020; i.e., 14 WYs) datasets (Figure 12). Here, we used parameters calibrated with the five representative OFs and the TUW model, using data from the period April/1994 – March/2013. The results show that the VE-Sep remains the top-performing objective function in terms of CRPSS, while Split KGE yields the worst results. Further, the rankings of the other objective functions (NSE, ModKGE, and KGE(Q)+NSE(log(Q))) vary depending on the verification period, and CRPSS values are higher during the calibration period compared to the evaluation datasets. We have added this new figure (to complement Figure 11) and text (L445-451) in Section 5.5, following the reviewer's recommendation.

[Figure]

Figure 12. Comparison of CRPSS for September 1 hindcasts obtained with the five representative objective functions and the TUW model. Each panel contains results for a different hindcast verification period: (left) 33 WYs (full period); (middle) 19 WYs (calibration period); and (right) 14 WYs (i.e., evaluation dataset April/1987 – March/1994 and April/2013 – March/2020). Each boxplot comprises results from the 22 case study basins and one objective function. The boxes correspond to the interquartile range (IQR, i.e., 25th and 75th percentiles), the horizontal line in each box is the median, and whiskers extend to the ±1.5·IQR of the ensemble. The numbers in parentheses denote the median CRPSS among all basins, and the numbers above the OF ranking based on that median, being 1 the best.

*Other comments:*
*Why were elevation bands defined only for CemaNeige?*

We did not use snow elevation bands in SAC/SNOW 17 and TUW models because preliminary experiments showed that the benefits of adding these on the KGE of daily flows were marginal. We stress that the use of three models does not seek to provide comparisons among different model structures; instead, we aim to examine to what degree our results and conclusions can be model-dependent. We have made these points in L182-186:

"While the CemaNeige is configured with 10 elevation bands, the snow routines of TUW and SAC-SMA (i.e., SNOW-17) are implemented in a lumped fashion because preliminary experiments with these models showed that the benefits of adding snow bands on the KGE of daily flows were marginal. We stress that the use of three models does not seek to provide comparisons among different model structures; instead, we aim to examine to what degree our results and conclusions can be model-dependent."

*Although I did not mention it in the first report, I find Figure 10 difficult to understand. In order to better support the main conclusions related to Section 4.4, I suggest that the authors reduce the content of this figure. For example, only the results for two initialisation times and TUWmodel could be shown in this figure.*

We have modified this Figure to show the results for the TUW model, following the reviewer's recommendation. We have decided to keep the five initialization times to illustrate the progression of the relationship between hindcast performance and catchment attributes, and the results for the SAC and GR4J models are included in the supplements (Figure S12). Accordingly, we have made some re-organization and modifications on the text (L342-355):

"We now explore the factors that control seasonal hindcast quality, and the extent to which the choice of calibration metric impacts the connections inferred from our sample of catchments. Figure displays results for the TUW model only, and the full results (including GR4J and SAC) are available in the Supplement. In general, the choice of calibration metric affects more the strength, rather than the sign,

of the relationships between hindcast quality and catchment attributes. In particular, we find that the correlations between CRPSS and catchment descriptors obtained with Split KGE (which maximizes hydrologic consistency), are weaker than those obtained with other calibration metrics (e.g., see results for baseflow index with TUW, interannual runoff variability with all models, and fraction of precipitation falling as snow with all models).

We find statistically significant correlations between CRPSS and the baseflow index ($\rho \sim 0.2 – 0.8$) with the three models, being ModKGE ($\rho = 0.49$), VE-Sep ($\rho = 0.70$), and VE-Sep ($\rho = 0.41$) the objective functions that maximize such relationship for September 1 when using TUW (Figure), GR4J and SAC (Figure S12), respectively. Figure shows significant correlations between CRPSS and the interannual variability of runoff ($\rho \sim 0.0 – 0.6$) – especially for September 1 hindcasts ($\rho = 0.53$ for VE-Sep/TUW, $\rho = 0.64$ for ModKGE/GR4J and $\rho = 0.62$ for VE-Sep/SAC). Also positive, but generally weaker correlations are obtained between hindcast skill and p-seasonality ($\rho \sim -0.6 – 0.0$), as well as the fraction of precipitation falling as snow ($\rho \sim 0.0 – 0.4$)."

[Figure]

Figure 10. Spearman's rank correlation coefficients between catchment characteristics (shown in different rows) and the CRPSS (left), α reliability index (center), and the coefficient of determination R2 (right) obtained for seasonal streamflow hindcasts (period April/1987 – March/2020), produced with the five representative objective functions (x-axis in each color matrix), different initialization times (y-axis in each color matrix), and the TUW model. Black dots indicate statistically significant ($p < 0.05$) correlations.

*Out of curiosity, I looked at your scripts and saw that you tested GR5J and GR6J. Was there a reason for keeping only GR4J? Does using GR6J solve the problem with the X4 parameter (unit hydrograph*

*time constant which is often close to the upper limit for almost half of the catchments after calibration, probably when baseflow is high)?*

Preliminary experiments with the GR5J and GR6J models (Figure R1) – reported by Araya (2022) – provided worse results in some basins compared to GR4J using the KGE as objective function, especially during the period 2013-2020 (Figure R1c). Even though GR6J addresses the issue with the X4 parameter (whose upper limit is 20) in some basins, it still provides X4 values near the upper limit in some catchments. In fact, X4 > 19 with the GR6J (GR4J) model in 9 (7) out of 22 basins.

[Figure]

Figure R1. KGE computed with daily flows for all basins, obtained from calibrations with GR4J, GR5J, GR6J (all including CemaNeige) and TUW models using KGE as the objective function. The results are displayed for the periods (a) 1994-2013, (b) 1987-1994, and (c) 2013-2020.

Table R1. Parameter values obtained from calibrating GR4J and GR6J (both including CemaNeige), using KGE as the objective function.

| BNA Basin | $X_4$ GR4J | $X_4$ GR6J | BNA Basin | $X_4$ GR4J | $X_4$ GR6J |
|---|---|---|---|---|---|
| 3414001 | 14.45 | 19.49 | 5200001 | 19.91 | 12.74 |
| 4302001 | 15.24 | 5.29 | 5410002 | 20.00 | 20.00 |
| 4301002 | 15.58 | 19.97 | 5411001 | 1.03 | 1.03 |
| 4311001 | 20.00 | 16.26 | 5722001 | 20.00 | 15.34 |
| 4501001 | 8.30 | 3.32 | 5721001 | 20.00 | 19.99 |
| 4522002 | 1.62 | 1.61 | 5710001 | 17.72 | 19.89 |
| 4511002 | 20.00 | 20.00 | 6027001 | 1.14 | 1.51 |
| 4513001 | 1.60 | 19.53 | 7103001 | 1.02 | 1.36 |
| 4703002 | 20.00 | 20.00 | 7112001 | 1.04 | 1.31 |
| 5101001 | 1.00 | 1.13 | 7115001 | 1.40 | 1.44 |
| 5100001 | 1.63 | 19.47 | 8104001 | 1.00 | 1.30 |

*L197: why does the model evaluation also cover part of the spin-up period between 1987 and 1994? Perhaps I have missed an explanation somewhere.*

We have modified the text to clarify this point. Regarding the spin-up period use in model calibration (L189-191):

"To compute the calibration objective function, we use modeled and observed streamflow data from the period April/1994 – March/2013 because it spans a diverse range of hydroclimatic conditions, considering the period April/1986 – March/1994 for model spin-up."

Regarding the spin-up period used to produce runoff simulations for the evaluation dataset (L197-201):

"Model evaluation is conducted by computing performance metrics with data from two periods: (i) April/1987 - March/1994, which is hydroclimatically diverse, and (ii) April/2013 – March/2020, which is characterized by unprecedented and temporally persistent dry conditions (Garreaud et al., 2017, 2019). To produce runoff simulations for each period, the preceding eight years (i.e., April/1979 - March/1987 and April/2005 – March/2013) were used for model spin-up."

*Figure 5: increase size by rearranging panels.*

We have increased the panel size by reducing the horizontal scale of the figure and, and we have adjusted the y-axes scale to enlarge the boxes. Additionally, we have compressed the panels to use the space more efficiently.

[Figure]

Figure 5. Time series with ESP seasonal hindcasts (i.e., September-March runoff) initialized on July 1 (left panels), and September 1 (right panels) for the Maipo at El Manzano basin. The boxes correspond to the interquartile range (IQR, i.e., 25th and 75th percentiles); the horizontal line in each box is the median, whiskers extend to the ±1.5·IQR of the ensemble, and the red dots represent the observations. The results were produced with the TUW model, using parameters obtained from calibrations conducted with NSE, KGE(Q)+NSE(log(Q)) and VE-Sep (see details in Section 3.1). Each panel displays the CRPSS, the reliability index α, and the coefficient of determination $R^2$ (computed using the ensemble hindcast median).

*Figure 9: use the same scale for the y-axis. Why is "Split KGE" in second place for TUW? The median of the average bias in hydrological signatures seems to be lower than the median of KGE+NSE(log). Have you ranked the objective functions after rounding up the variable?*

We now use the same scale for the y-axis of the panels that belong to the same row.

[Figure]

Figure 9. Variations in September 1 CRPSS due to the choice of popular and alternative objective functions (shown in different boxplots), relative to the best performing OF in terms of forecast quality (VE-Sep). The dashed line indicates no difference (i.e., no loss) in forecast performance. The bottom panel display the average bias in hydrological signatures (computed over the calibration and evaluation periods) with the associated ranking (being 1 the best in terms of hydrological consistency), and median average bias obtained from the sample of basins (in parentheses).

Split KGE is in second place for TUW because its median is 20.004%, while the median for KGE+NSE(log) is 19.988%. As expected, such small difference is very hard to see, even after zooming in (see attached figure). Note that the metrics were not rounded up to rank the objective functions.

[Figure]

*Figure S1: why is the period covered here only between 2014 and 2016?*

Figure S1 was created to complement the results of Figure 4, which shows simulated and observed daily hydrographs for the period April/2014 - March/2016, and runoff seasonality for the evaluation dataset (April/1987 – March/1994 and April/2013 – March/2020) at the Maipo at El Manzano River basin, along with performance metrics. In the manuscript, it is stated, 'Similar results are obtained [...] for the remaining basins…' which is supported by Figure S1, which shows the daily KGE and the coefficient of determination between simulated and observed mean monthly runoff, for all 22 basins and the same periods examined in Figure 4. Hence, Figure S1 displays daily KGE for the period April/2014 - March/2016 to maintain consistency with Figure 4.

We have modified the caption of Figure S1 to clarify this:
"Kling-Gupta Efficiency (KGE) between simulated and observed daily streamflow for the period

April/2014-March/2016, using parameter values obtained with different calibration objective functions and hydrological models (upper panel); and coefficient of determination ($R^2$) between mean monthly simulated and observed runoff averages for the evaluation dataset (April/1987 – March/1994 and April/2013 – March/2020, from evaluation period (bottom panel). Each boxplot comprises results from the 22 case study basins. The boxes correspond to the interquartile range (IQR, i.e., 25th and 75th percentiles), the horizontal line in each box is the median, and the whiskers extend to the ±1.5·IQR of the ensemble. The points correspond to outliers beyond the whiskers' range."

**References**

Araya, D. (2022). Evaluación de la metodología ESP para la generación de pronósticos de caudales de deshielo en cuencas de Chile Central (in Spanish). Available at:
https://repositorio.uchile.cl/handle/2250/185501